# Development of A Global 5arcmin Groundwater Model (H08-GMv1.0): Model Setup and Steady-State Simulation

Qing He<sup>1</sup>, Naota Hanasaki<sup>1,2</sup>, Akiko Matsumura<sup>3</sup>, Edwin H. Sutanudjaja<sup>4</sup>, Taikan Oki<sup>1</sup>

Correspondence to: Qing He (heq16@tsinghua.org.cn)

**Abstract.** Groundwater plays a critical role in regulating the global hydrological cycle and serves as the most stable freshwater resource for human daily water consumption. However, many global water models, including H08, a global water model considering human water use activities, downplay the groundwater component, i.e., the underground aquifer is often described as a simple lumped model where no lateral groundwater movement or the water table is represented. Here, we present a global H08-MODFLOW groundwater model (H08-GM), built at a five-arcmin spatial resolution, aiming to enhance the capability of the original H08 model in simulating groundwater flows. We describe the basic model setups and simulations under steadystate conditions in this paper. The Local One-At-A-Time (OAT) Sensitivity Tests are first conducted to select the best-run model simulations against in-situ observations. At the global scale, all model runs demonstrate overall good performance of groundwater head, whereas perform poorly in simulating Water Table Depth (WTD, groundwater table below land surface), which is shown to be a common issue in other global groundwater models. Our analysis also reveals two complementary global relationships: one between groundwater depth and topographic slope, and another along gradients of human activity (irrigation and population), together demonstrating how natural and anthropogenic processes jointly control the spatial distribution of WTD. We further use the model to reveal the mechanisms controlling groundwater flow dynamics and present the global cellto-cell net groundwater lateral flow map. We found that the magnitude of the net groundwater lateral flow in some regions is non-negligible to annual groundwater recharge. This highlights the important role of the lateral groundwater flow in maintaining the regional water budget. The steady-state simulation from this study provides the necessary initial condition for the transient simulations, which is essentially important to analyze the global groundwater decline trends and will be presented in another paper. Although developed in the one-way coupled manner, the H08-GM model can provide a powerful tool for large-scale groundwater studies, which enables direct comparison with other groundwater models joined the Inter-Sectoral Impact Model Intercomparison Project (ISIMIP), and is essential to advance the development of the next-generation global water models.

<sup>&</sup>lt;sup>1</sup>Department of Civil Engineering, The University of Tokyo, Tokyo, 113-8656, Japan

<sup>&</sup>lt;sup>2</sup>National Institute for Environmental Sciences, Tsukuba, 305-8506, Japan

<sup>&</sup>lt;sup>3</sup>Nippon Koei Co. Ltd., Tsukuba, 305-0047, Japan

<sup>&</sup>lt;sup>4</sup>Department of Physical Geography, Faculty of Geosciences, Utrecht University, Utrecht, the Netherlands

### 30 1 Introduction

Groundwater plays a critical role in the global hydrological cycle. The water exchange between aquifers and surface water bodies buffers the sharp seasonal fluctuations in river channels and lakes, maintaining the resilience of aquatic landscapes and ecosystems (Huggins et al., 2023; Jasechko et al., 2021; Otoo et al., 2025; Rohde et al., 2024a, b; Saccò et al., 2024). Such surface-groundwater exchange can also contribute to a significant amount of rainfall and evapotranspiration variability in arid and semi-arid regions (Bierkens and Van Den Hurk, 2007; Condon and Maxwell, 2019; Schaller and Fan, 2009), therefore mitigating the severity of droughts and heatwaves through land-atmosphere interactions (Keune et al., 2016; Kollet and Maxwell, 2008; Maxwell et al., 2007; Mu et al., 2022).

Groundwater serves as a natural freshwater reservoir to supply human water use activities. Due to its large storage capacity and slow flow rate, groundwater contributes as the major and the most stable freshwater source to human water use in households, agriculture, and industry (Gleeson et al., 2012; Kuang et al., 2024; Medici et al., 2024; Mukate et al., 2020; Scanlon et al., 2023; Wada et al., 2014). On a global average, more than 90% of freshwater availability excluding glaciers is contributed by groundwater storage (Margat and Gun, 2013). In extremely arid regions where no surface water is available, or during dry seasons when no rainfall recharges surface water bodies, groundwater could be the only water source for the local communities (Braune and Xu, 2010; Calow et al., 2010; Gee and Hillel, 1988). Therefore, understanding the spatial and temporal distribution of groundwater availability is key to addressing water scarcity at local, regional, and global scales.

Global Water Models (GWMs) provide useful tools to understand the role of groundwater in terrestrial water cycle (Reinecke et al., 2025). However, at the early stage of GWMs development, the groundwater processes are often downplayed due to the computational resource limitation. For example, many GWMs such as WaterGAP (Döll et al., 2003), PCR-GLOBWB (van Beek et al., 2011), H08 (Hanasaki et al., 2008a, b), and CLM (Dai et al., 2003), etc., chose to simplify the aquifer as a bucket reservoir and only represent the vertical water exchanges. In the real world, the groundwater flows three-dimensionally, including both vertical flux exchanges with the upper unsaturated zones, and horizontal flows from areas of high hydraulic head to adjacent low-head regions. The groundwater lateral flows are proven to contribute a substantial amount to the total natural water budget, especially in high spatial resolution studies (Akhter et al., 2025; Krakauer et al., 2014; Miguez-Macho and Fan, 2025), and in regions of groundwater convergence and arid climates (de Graaf and Stahl, 2022). Such simplification could introduce considerable bias to the models' estimation of total water availability. The absence of explicit representation of the groundwater table also undermines the hydrological models' capability for direct and accurate evaluation of human water withdrawal impact on groundwater depletion, particularly over intensively exploited regions such as Ogallala Aquifer and North China Plain (Cao et al., 2013; Scanlon et al., 2012; Yang et al., 2022).

With the advancements in computational technologies, the representation of lateral groundwater flow has re-invoked interest from the GWM communities in recent two decades (Condon et al., 2021; Gleeson et al., 2021). Among them, benchmark efforts have been made by the PCR-GLOBWB group (de Graaf et al., 2015, 2017; Sutanudjaja et al., 2011, 2018; Verkaik et al., 2024), where the original bucket groundwater module has been replaced by MODFLOW, a physical groundwater model with 3-Dimensional flowing processes based on Darcy's Law (Harbaugh, 2005; Harbaugh et al., 2000; Langevin et al., 2017; McDonald and Harbaugh, 1988). Noteworthy efforts to address the lateral groundwater flow issues in GWMs are also seen in WaterGAP 2.0 (Müller Schmied et al., 2021; Reinecke et al., 2019a), where a gradient-based global groundwater flow parameterization scheme has been developed and implemented; The development of 3-Dimensional saturate flow module in MATSIRO (Koirala et al., 2014); The coupling of ParFlow to CLM (Maxwell et al., 2015; Maxwell and Miller, 2005); and a newly developed hydro-economic model CWatM (Burek et al., 2020; Guillaumot et al., 2022) (The list of models here is illustrative, not exhaustive). With explicit lateral flow processes and groundwater table represented, the current generation of GWMs is now able to estimate decadal groundwater storage changes and groundwater level declines caused by human water pumping activities. This advancement enables the direct comparison with the observation and estimations from data-driven approaches (Kuang et al., 2024; Scanlon et al., 2023).

Here, we present a global H08-MODFLOW model (H08-GM hereafter) to better represent groundwater lateral flow, thereby improving the realism of simulated groundwater availability and human-groundwater interactions in the original H08. We will first describe the basic model setups, including the coupling framework, parameterization schemes, and the hydrogeological data and in-situ validation data used in this paper. The global 41-year (1979-2019) steady-state simulation (i.e., time was removed from the model formulation rather than using a transient simulation to reach an equilibrium) results under pristine conditions (i.e., without human groundwater pumping), mainly the spatial distribution of the climatological groundwater depth and aquifer-river channel water flux exchange regime will be included in this study. The steady-state simulation is useful for understanding the long-term balance between recharge and discharge, and provides initial conditions to the transient simulation, which will be discussed in another study. The development of the H08-GM model will allow direct assessments on how lateral flow from adjacent areas can mitigate groundwater decline in highly exploited aquifers, thus aiding in the evaluation of global water scarcity and informing water management strategies. The explicit representation of the groundwater table in the H08-GM model will also facilitate a more accurate comparison with outputs from other GWMs that joined the Inter-Sectoral Impact Model Intercomparison Project (ISIMIP).

### 90 2 Data and Methods

### 2.1 General Description: The Coupling Framework

The H08-GM consists of two parts: the surface water processes simulated by H08, and the groundwater processes simulated by MODFLOW. In this section, we provide an overview of how the two models can be connected. Detailed descriptions of the individual models will be given in Sections 2.2 and 2.3, respectively.

An illustration of the groundwater hydrology components (land surface elevation (Elv), groundwater head (Head), aquifer bottom elevation and thickness, river—aquifer interaction, and pumping zones) is shown in Figure 1b to help better understand the terminologies throughout the paper. The conceptual framework of the coupled model is shown in Figure 1a. The two models are connected through the water flux exchanges, i.e., at each model grid H08 provides total groundwater recharge, river discharge, and total groundwater withdrawal rate as the hydrological forcing to MODFLOW (red arrows, Figure 1a); Baseflow and groundwater level are then simulated by MODFLOW and outputted back to H08 (grey arrow, Figure 1a). An I/O interface that can store the output of H08 and MODFLOW is essential to achieve the two-way coupling goal, with functions of: (i) Time keeping; (ii) Variables and units converting (e.g., from groundwater levels to storage); and (iii) Exchange prognostic/diagnostic variables. From H08 to MODFLOW, the spatially distributed recharge, river stage/flow will be passed; From MODFLOW to H08, the groundwater heads, simulated baseflow to river channels, and root-zone capillary rise fluxes, which feedback to H08 evapotranspiration stress and to the dynamic water-allocation module. However, as an initial step, in this study we only present the offline simulation results (i.e., no feedback from MODFLOW to H08), in order to test whether the forcing from H08 can produce reasonable global groundwater simulation by driving MODFLOW. Both models are built on a 5 arcmin grid to ensure consistent spatial resolution. All the land surface variables in H08 relevant to MODFLOW model (e.g., recharge, runoff, etc.) are simulated at the flux density level (i.e., no grid cell area is involved). Therefore, we did not apply areal and volumetric fluxes adjustment here.

### 2.2 Surface Water Model H08

The H08 model is a global hydrological model considering human water use activities (Hanasaki et al., 2008a, b, 2018). The model considers natural hydrological processes maintaining a closed energy and water balance at each model grid. The soil column is described as a one-layer leaky bucket with a fixed depth of 1m and water draining consecutively at the bottom (subsurface runoff). Soil moisture is obtained through the water balance equation, considering rainfall, snowmelt, evapotranspiration, surface and subsurface runoff, and groundwater baseflow. Evapotranspiration is calculated linearly to the potential evapotranspiration based on a stress factor considering soil moisture. Surface runoff is described as the residual water exceeding soil capacity, while subsurface runoff is calculated as a power function of soil moisture. River discharge is accumulated from surface runoff by the river routing module at each grid. All grid cells within each Köppen climate zone share uniform parameter settings (e.g., soil wilting point and field capacity). Although there is no subgrid distinction between

vegetated or bare soil fractions, neither is the soil capillary rise characterized in H08, the overall simulated hydrological regimes correspond reasonably well to the Budyko aridity framework on a global average, i.e., soil moisture dominates evapotranspiration in arid areas, while net radiation dominates in humid areas (Hanasaki et al., 2008a). Human water infrastructures, including reservoir operations, desalination plants, and inter-basin water transfer through aqueducts and canals, are also available options based on the users' purposes.

Figure 1. (a) Schematic diagram for H08-GM framework. The upper part of the raster represents the natural hydrological processes and human water withdrawal for different sectors in H08. The lower part of the raster represents groundwater processes. The red arrow indicates hydrological forcing input from surface water (H08) to a single-layer groundwater aquifer with structured grid (MODFLOW). The grey arrow indicates groundwater feedbacks to surface water. In the current model setting, only the red arrow part is enabled (one-way coupling).

(b) Schematic diagram of groundwater hydrology components. Yellow triangles represent the phreatic surface. The difference between surface elevation and groundwater head is termed as Water Table Depth (WTD).

The groundwater aquifer is described as a single-layer reservoir, where the groundwater storage is fed with groundwater recharge ( $Q_{rc}$ , Eq. (1)) calculated proportionally to the total runoff. There is no characterization on the aquifer geometry; Only the storage changes are available. The groundwater discharge (baseflow) is calculated as a power function of the groundwater storage. Two types of aquifers are introduced: renewable and non-renewable. The renewable aquifers can receive water from groundwater recharge, whereas in the non-renewable aquifer, water can only be withdrawn but not replenished. Human water withdrawal is used for three sectors, i.e., household, industry, and agriculture. The total extracted water for agriculture is calculated dynamically based on irrigation water requirement during crop growth, while water withdrawal for the other three sectors is calculated based on the static sectoral water requirement maps provided by AQUASTAT (Food and Agriculture Organization, 2007). The fraction of groundwater use per sector per country from International Groundwater Resources Assessment Centre (IGRAC) database (IGRAC, 2004) is used to determine how much water is abstracted from surface water bodies and how much is from groundwater aquifers. Water abstraction from renewable aquifers has a higher priority than the non-renewable ones. For brevity we only summarize the key elements relevant to this study here, more details are referred to (Hanasaki et al., 2008a) and (Hanasaki et al., 2018).

$$Q_{rc} = \min \left( Q_{rc_{max'}} f_r \cdot f_t \cdot f_h \cdot f_{pq} \cdot Q_{tot} \right), \tag{1}$$

Where,  $Q_{rc_{max}}$  is the maximum groundwater recharge (kg m<sup>-2</sup> s<sup>-1</sup>),  $f_r$  is a relief-related factor (0 <  $f_r$  < 1),  $f_t$  is a soil-texture-related factor (0 <  $f_r$  < 1),  $f_t$  is a hydrogeologyrelated factor (0 <  $f_h$  < 1),  $f_{pg}$  is a permafrost/glacierrelated factor (0 < fpg < 1), and  $Q_{tot}$  is the total runoff (kg m<sup>-2</sup> s<sup>-1</sup>).  $Q_{rc_{max}}$ ,  $f_r$ ,  $f_t$ ,  $f_h$  and  $f_{pg}$  are determined by the look-up tables provided in Tables A1–A4 of Döll and Fiedler et al. (2008).

**Figure 2**. Long-term averaged groundwater recharge and river discharge from H08 (1979-2019). (a) Global distribution of the 41-year averaged groundwater recharge (unit: mm  $d^{-1}$ ). (b) - (d) for the 41-year mean river discharge (unit:  $m^3$  s<sup>-1</sup>) in the southern Mississippi River basin (b), the India Peninsula (c), and the Yellow and the Yangtze River basins in China (d), respectively.

We first run H08 individually to obtain the groundwater recharge and river discharge to drive MODFLOW. Global meteorological forcing data including 8 variables, i.e., rainfall, 2 m air temperature, specific humidity, wind speed, surface air pressure, longwave and shortwave radiation, and snowfall, from the W5E5 dataset (Lange et al., 2021) are used. The W5E5

dataset was compiled based on version 2.0 of WATCH Forcing Data methodology applied to ERA5 data (WFDE5) (Cucchi et al., 2020; Weedon et al., 2014), ERA5 reanalysis data (Hersbach et al., 2020), and precipitation data from version 2.3 of the Global Precipitation Climatology Project (GPCP) (Adler et al., 2003). The WFDEI data is originally at 0.5 degrees and was post-processed to a 5 arcmin resolution using the linear interpolation function embedded in H08, i.e., the values of the four surrounding grid cells for a certain grid cell will be used to calculate a linear interpolated value by weighting each using the distance ratio. The model was run under the natural scenario at the monthly timestep from January 1, 1979 to December 31, 2019. For the steady-state simulation described in this paper, only groundwater recharge and river discharge are used and averaged over the simulation period to obtain the 40-year climatological means. The spatial distributions of these two variables are shown in Figure 2.

Groundwater recharge (Figure 2a) is generally higher in humid regions (e.g., the eastern United States, Europe, and southern China) and lower in arid regions (e.g., the western United States, Arabian Peninsula, and inland Eurasia), with maximum values observed in tropical areas such as the Amazon Basin, the Sahel, and the Indonesian archipelago. Figure 2(b)–2(d) shows the spatial distribution of river discharge across representative basins in the United States (lower Mississippi), India (Ganges–Brahmaputra), and China (Yangtze and Yellow Rivers). All three regions show a consistent pattern – the high discharge is mainly concentrated along major rivers and downstream reaches and lower values in upstream or small tributaries. The discharge values are also notably lower under drier climates (Yellow River) compared to other basins under more humid hydroclimatological conditions.

### 2.3 Groundwater Model MODFLOW

MODFLOW is the USGS's modular hydrologic model for simulating and predicting groundwater conditions (Langevin et al., 2017; McDonald and Harbaugh, 1988). The model uses a generalized control-volume finite-difference approach to solve the two- and three-dimensional groundwater flows based on Darcy's equation. Lateral flows and groundwater heads are explicitly simulated and provided as outputs. The modular structure also allows users to customize the model flexibly by adding packages of their research targets such as aquifer properties, recharge, rivers, and wells, etc. In this study, we use MODFLOW6 (version 6.4.0), to build a global single-layer unconfined groundwater model and replace the original groundwater store in H08. FloPy (Bakker et al., 2016) (version 3.3.6) is used as the interface to run the model.

# 190 2.3.1 Aquifer Properties

185

195

Two aquifer property parameters, i.e., aquifer thickness and hydraulic conductivity, are required to build an unconfined groundwater steady-state model at any spatial scale. Aquifer thickness refers to the vertical extent between the top and bottom boundaries of the aquifer (Figure 1b); For a given area, it indicates the aquifer's potential water storage capacity. This parameter is usually obtained from field experiments in local scale studies, while the global map is often delineated based on lithological categories. The GLobal HYdrogeology MaPS (GLHYMPS) (Gleeson et al., 2011, 2014; Huscroft et al., 2018) is

one pioneering dataset of such a type. However, the GLHYMPS aquifer thickness only accounts for the shallow layer (thickness up to 100m), thus cannot reasonably represent the deep aquifers in the world. A terrain-based approach was then proposed by (de Graaf et al., 2015) and shown to be effective for deep aquifer characterization based on the calibration of transmissivity to observed heads. The hypothesis of this approach is that there is similarity in Coefficient of Variations (CV) of aquifer thickness all around the world. Therefore, it first generates a random distribution of the average aquifer thickness based on the land surface and floodplain elevation differences ( $\Delta z$ ) at each grid. Then, observed statistical values from 6 regional scale studies are used to constrain the corresponding log-normal transformation of  $\Delta z$  and a standard normal ordinate function (i.e.,  $\varphi(z) = \frac{1}{\sqrt{2\pi}}e^{-z^2/2}$ ). The optimal guess is then derived as the final aquifer thickness product. In this study, we use this product to better represent the deep aquifers. The aquifer thickness map is shown in Figure 3(a).

205

200

Aquifer Hydraulic Conductivity [m  $d^{-1}$ ]

**Figure 3**. Global distribution of aquifer thickness (a) and aquifer hydraulic conductivity (b). The aquifer thickness product is from de Graaf et al. (2015)(de Graaf et al., 2015), and aquifer hydraulic conductivity is based on GLiM lithological map (Hartmann and Moosdorf, 2012) and GLHYMPS (Gleeson et al., 2011, 2014)

Hydraulic conductivity controls the rate at which groundwater flows through the aquifer materials and is primarily determined by the aquifer's lithological characteristics. The gridded 5arcmin GLiM global lithological map (Hartmann and Moosdorf, 2012) is used to define the spatial distribution of 16 lithologies (Figure C1, Appendix). For each lithological category, we obtain the corresponding permeability value from GLHYMPS; when there are multiple values (for subcategories) within one lithology type, we take their means based on the subcategory sample numbers. The standard deviation for each category is also obtained for the following sensitivity analysis in Section 2.4. The aggregated permeability data for each lithological type is shown in Table B1 (Appendix). The permeability is then converted to hydraulic conductivity as a direct model input. For permafrost regions (i.e., permafrost zonation index>0.5) (Gruber, 2012), we reduce K by one-order-of-magnitude by considering the combined effects of soil temperature, soil texture and freeze-thaw dynamics (Watanabe and Flury, 2008; Watanabe and Osada, 2016), although we note this is a strong assumption to ensure the model's numerical stability.

## 2.3.2 River Channel Properties (position, level, bottom elevation, riverbed drainage conductivity)

220

To investigate the river-aquifer exchanges, a river package (RIV) is used. The water flux exchanges are calculated based on the head difference between river channels and the aquifer cells, i.e., water leaks from the river channel to the aquifer when the river water level is higher than the groundwater head and vice versa, as:

$$Q_{flux} = c_{rb} \times (H_{riv} - h_{aq}) \tag{2}$$

Where,  $H_{riv}$  and  $h_{aq}$  refers to river water level (m) and groundwater head (m), respectively. When the groundwater table is below the river bottom, river bottom elevation ( $R_{bot}$ ) is used for  $h_{aq}$  to limit the maximum water flux exchanges.  $c_{rb}$  indicates the riverbed conductance ( $m^2d^{-1}$ ) and is calculated as:

$$c_{rb} = \frac{RIV_{wth} \times RIV_{len}}{r_{rb}} \tag{3}$$

Where,  $RIV_{wth}$  and  $RIV_{len}$  are the river channel width (m) and length (m), respectively, both of which are taken from (Yamazaki et al., 2011).  $r_{rb}$  is the riverbed resistance. In de Graaf et al. (2015), it is taken as 1 day. However, in our preliminary analyses we found the simulated head is rather sensitive to this parameter. Therefore, the appropriate value will be selected from several sensitivity experiments. See Section 2.4 for a detailed description.

Figure 4. Illustration of the groundwater model grid designations and river properties over the southern Mississippi river basin. (a) Schematic diagram of the river channel geometry.  $H_{riv}$  represents the river water level (unit: m) which serves as the input to calculate river-aquifer exchange; (b) Spatial distribution of river width (unit: km) from GWD-LR product (Yamazaki et al., 2014). Data over lake areas are not available; (c) Model designation of MODFLOW. Black, orange, and blue color represents river, drainage, and constant head grids, respectively; (d) Spatial distribution of river water levels (unit: m) calculated from Eq. (4).

We first use a combined satellite and empirical algorithm river width product GWD-LR to allocate the river grids in MODFLOW (Yamazaki et al., 2014). This product was constructed by applying the SRTM Water Body Database (SWBD) and the HydroSHEDS flow direction map, and shows high realism in representing river width for large river channels. To overcome its limitation in representing small rivers and overestimation of large rivers, we further constrained the results by

applying a power-law algorithm, as done in the latest version (v4.20) of the Hydrodynamic flood model CaMa-Flood (Yamazaki et al., 2011). See further description in Text S1 in Supplementary Materials. Because the river-aquifer exchange can be negligible for small tributaries, we define river width larger than 10 m as river grids where water exchanges actually happen, similar to the criteria defined in de Graaf et al. (2015). An illustration of the river width result and the resulting river grid allocation in MODFLOW are shown in Figure 4(b) and (d).

250

260

270

275

Next,  $R_{bot}$  is calculated as the difference between land surface elevation (DEM) and river channel depth ( $D_{chn}$ ) (Figure 4(a)), where the previous is taken from 30 arcsec HydroSHEDS dataset (Lehner et al., 2008) and aggregated to 5arcmin resolution using simple linear interpolation algorithm.  $D_{chn}$  is calculated based on the power-law algorithm as in CaMa-Flood model (Yamazaki et al. 2011) (Text S1 in Supplementary Materials).  $H_{riv}$  is then calculated as:

$$H_{riv} = R_{bot} + D_{riv} \tag{4}$$

Where,  $D_{riv}$  is the river water depth (m) and is calculated based on Manning's equation:

$$D_{riv} = \left(\frac{n \times Q_{chn}}{RIV_{wth} \times RIV_{slb}^{0.5}}\right)^{0.6} \tag{5}$$

Where, n is the Manning roughness coefficient and is taken as  $0.035 \,\mathrm{m}^{-1/3}\mathrm{s}^{-1}$ .  $RIV_{slp}$  refers to river channel slope (unitless) and is calculated as the ratio of the DEM difference between the current and next downstream river cells over the distance between the two cells. See (Oki and Sud, 1998) and (Yamazaki et al., 2009) for complete explanations about how the flow direction is decided and the distance between one cell and the next downstream cell is calculated.  $Q_{chn}$  refers to the river discharge (m³/s). For the steady-state simulation in this study, it is calculated as 40yr mean of the monthly H08 simulation. See Figure 2 (b) – (d) for examples of the spatial distribution over southern Mississippi river basin, Indian Peninsula, and Yellow and Yangtze River basin in China.

### 265 2.3.3 Other Boundary Conditions (constant head, topography, drainage)

Unlike H08, MODFLOW requires land surface elevation data to calculate groundwater movement. We use DEM from HydroSHEDS for this purpose. For all ocean grids, since the submarine flow is not our research focus, we set them as constant head (CHD) with the water level of 0 m, i.e., it can receive (release) unlimited water from (to) the terrestrial underground aquifer. We also do not separately consider evapotranspiration in the groundwater model because it is already included in the H08 simulation part. For small tributaries (river sequence number less than 10), since the water entering the aquifer system can be negligible, we apply the drainage package (DRN) to allow water to leave the groundwater system. When  $h_{aq}$  is above a prescribed level, here set as DEM, water from the groundwater will form ponding areas and be removed from the aquifer system. The drainage rate is calculated based on land surface water conductance, calculated in the same form as Equation (2). No water flux exchange will happen when  $h_{aq}$  is below the drainage level (DEM). The allocation for DRN and CHD grids in MODFLOW is illustrated over an example region in Figure 4(c).

Table 1. Sensitivity experiment setting scenarios and the resulting groundwater head simulation statistics against observations. 
† Indicates the best-run experiment; ref for K indicates the mean hydrological conductivity from Gleeson et al. (2011); ref for Rch indicates the 40-year mean H08 recharge; ref for  $r_{rb}$  indicates 1 day. For the model and observation difference terms  $D_{mean}$  and  $D_{med}$ , positive values indicate the overall model head is shallower than observed head and vice versa.  $D_{std}$  is always positive; larger values indicate the overall simulated head deviates further from the observation.

|              | EXP                 | K          | Rch          | $r_{rb}$      | $R_{cor}$ | $D_{mean}$ | $D_{med}$ | $D_{std}$ |
|--------------|---------------------|------------|--------------|---------------|-----------|------------|-----------|-----------|
|              | K0R0B0              | ref        | ref          | ref           | 0.67      | 452.79     | 294.53    | 494.67    |
|              | K1R0B0              | $+1\sigma$ | ref          | ref           | 0.79      | 271.47     | 151.6     | 378.48    |
|              | K2R0B0              | $+2\sigma$ | ref          | ref           | 0.88      | 34.29      | 33.14     | 255.29    |
|              | K0R1B0              | ref        | $+0.5\sigma$ | ref           | 0.53      | 657.64     | 445.97    | 679.98    |
| A            | K1R1B0              | $+1\sigma$ | $+0.5\sigma$ | ref           | 0.70      | 397.89     | 229.16    | 505.27    |
|              | K2R1B0              | $+2\sigma$ | $+0.5\sigma$ | ref           | 0.85      | 89.5       | 60.39     | 290.62    |
|              | K0R2B0              | ref        | $-0.5\sigma$ | ref           | 0.80      | 240.37     | 133.58    | 334.19    |
|              | K1R2B0              | $+1\sigma$ | $-0.5\sigma$ | ref           | 0.85      | 123.08     | 63.22     | 283.06    |
|              | K2R2B0              | $+2\sigma$ | $-0.5\sigma$ | ref           | 0.83      | -68.92     | 2.58      | 297.93    |
|              | K0R0B1              | ref        | ref          | ×0.1          | 0.95      | 90.81      | 65.27     | 168.24    |
|              | K1R0B1              | $+1\sigma$ | ref          | ×0.1          | 0.95      | 63.3       | 41.31     | 169.7     |
|              | K2R0B1              | $+2\sigma$ | ref          | ×0.1          | 0.93      | -23.99     | 5.71      | 200.46    |
|              | K0R1B1              | ref        | $+0.5\sigma$ | ×0.1          | 0.94      | 115.33     | 87.11     | 176.22    |
| В            | K1R1B1              | $+1\sigma$ | $+0.5\sigma$ | ×0.1          | 0.94      | 85.92      | 56.44     | 175.47    |
|              | K2R1B1              | $+2\sigma$ | $+0.5\sigma$ | ×0.1          | 0.93      | 1.24       | 13.09     | 189.78    |
|              | K0R2B1              | ref        | $-0.5\sigma$ | ×0.1          | 0.95      | 57.91      | 40.62     | 165.09    |
|              | K1R2B1              | $+1\sigma$ | $-0.5\sigma$ | ×0.1          | 0.94      | 20.71      | 22.24     | 179.68    |
|              | K2R2B1              | $+2\sigma$ | $-0.5\sigma$ | ×0.1          | 0.90      | -76.58     | -6.22     | 247.33    |
|              | K0R0B2              | ref        | ref          | ×0.01         | 0.95      | 47.82      | 23.73     | 161.55    |
|              | K1R0B2              | $+1\sigma$ | ref          | $\times 0.01$ | 0.95      | 30.97      | 17.35     | 162.61    |
|              | K2R0B2              | $+2\sigma$ | ref          | $\times 0.01$ | 0.93      | -31.18     | 0.98      | 193.91    |
|              | K0R1B2              | ref        | $+0.5\sigma$ | $\times 0.01$ | 0.95      | 51.18      | 26.86     | 162.27    |
| $\mathbf{C}$ | K1R1B2              | $+1\sigma$ | $+0.5\sigma$ | $\times 0.01$ | 0.95      | 37.94      | 21.06     | 161.89    |
|              | K2R1B2 <sup>†</sup> | $+2\sigma$ | $+0.5\sigma$ | $\times 0.01$ | 0.94      | -17.98     | 4.24      | 184.88    |
|              | K0R2B2              | ref        | $-0.5\sigma$ | ×0.01         | 0.95      | 36.95      | 19.35     | 161.82    |
|              | K1R2B2              | $+1\sigma$ | $-0.5\sigma$ | ×0.01         | 0.95      | 10.59      | 11.78     | 172.13    |
|              | K2R2B2              | $+2\sigma$ | $-0.5\sigma$ | ×0.01         | 0.92      | -51.92     | -4.33     | 211.04    |

### 2.4 Local One-At-A-Time (OAT) Sensitivity Tests

Since uncertainties in the groundwater recharge and key aquifer parameters (i.e., aquifer hydraulic conductivity and thickness) are reported to be high (Gleeson et al., 2011; Reinecke et al., 2019b, 2021; Wada et al., 2010), we conducted several sensitivity tests to ensure the robustness of the simulated steady-state groundwater head. Additionally, our preliminary analyses show that the river geometry parameters, such as riverbed resistance, can also play an important role in the resulting groundwater head simulation. Therefore, in total, we select 3 parameters, i.e., groundwater recharge *RCH*, aquifer hydraulic conductivity *K*, and

riverbed resistance  $r_{rb}$ , for sensitivity analyses. The aquifer thickness D is not considered explicitly here because MODFLOW actually applies aquifer transmissivity (KD) for simulation, therefore the effect can be implicitly reflected in the variation in K. We note that the analyses here are local One-at-A-Time (OAT) only and do not address interaction effects; they therefore fall short of full sensitivity analyses objectives (screening, ranking, mapping) (Pianosi et al., 2016). The more rigorous global sensitivity analyses, as in (Reinecke et al., 2019b), will be pursued in future investigations.

To maintain computational efficiency, for each parameter we did three sensitivity analyses. Together this results in 27 experiments in total (Table 1). We take K0R0B0 as the reference experiment (ref), and use Correlation Coefficient (*R*), Mean, Median, and Standard Deviation of the difference between simulation and observation (*Dmean*, *Dmedian*, *Dstd*) to evaluate the performance of each experiment against observations. For parameters of *K* and *RCH*, one and two standard deviations are added individually for each relevant experiment. The statistic for *K* is from Gleeson et al. (2011) directly, while for *RCH* it is calculated based on groundwater recharge from H08 monthly simulation output (1979.01 – 2019.12). Note that although the aquifer thickness data we use is for deep aquifers while Gleeson et al. (2011) only provides such information for the shallow, here we assume there is similarity in aquifer thickness statistics between the two layers, similar to the assumption in the derivation of the dataset we use. For *RRD*, because there is no global reference of how its statistics should look like, rather simplistic scale factors are applied, i.e., 0.1 day and 0.01 day are taken for different experiment settings.

### **305 2.5 Validation**

To validate the simulation results, we use the equilibrium water table level observations from (Fan et al., 2013). In total, this dataset comprises 1,603,781 WTD readings, along with their corresponding elevation and geographic information. We then average the observations within the same model grid cell to mitigate the influence of the point-grid scale gaps as much as possible. We evaluate both the groundwater head and WTD, since the previous provides a more physically meaningful metric fundamental to groundwater flow dynamics (de Graaf et al. 2015), and the latter is more directly relevant to human and ecosystem water accessibility (Reinecke et al., 2024). The global scale model performance is evaluated first; Then, we evaluate the model behaviours in terms of different irrigation intensity and population density. Here, the irrigation intensity is represented by the global 10km irrigation area fraction map from (Siebert et al., 2015), and the population density is aggregated from the 1km global population dataset of year 2020 (https://hub.worldpop.org/geodata/summary?id=80026). Observations with invalid elevation readings are excluded. The total number of aggregated observations is 75,386.

In addition to the direct comparison between the simulated WTD against observations, we also compare the functional relationship between known drivers of groundwater flow (e.g., climatic aridity and topography) and WTD (Gleeson et al., 2021; Gnann et al., 2023; Reinecke et al., 2024; Wagener et al., 2022). The climatic aridity is calculated as the ratio of potential evapotranspiration to precipitation (PET/P, or Aridity Index (AI)), based on Global Land Data Assimilation System (GLDAS) Noah Land Surface Model L4 dataset (Rodell et al., 2007). AI>1 indicates the water-limited regime where atmospheric water

demand is larger than precipitation supply (dry climate in general); whereas AI

Figure 5. Scatterplots of the simulated groundwater head under different parameter settings (Experiment group C). Inserted texts refer to statistics between model simulation (y-axis) and observation from Fan et al. (2013) (x-axis). R<sup>2</sup>, n, Dmean, Dmed, and Dstd refers to coefficient of determination, sample size, mean, median and standard deviation of the simulation-observation difference.

### 335 3 Results and Discussion

### 3.1 Validation of Simulated Groundwater Head And the Sensitivity to Hydrogeological Parameters

The statistics of groundwater head from each sensitivity experiment result against observations are shown in Table 1 and Figure 5. The simulation-observation correlation coefficient of groundwater head ranges between 0.66 and 0.95 across experiments (p < 0.01), suggesting our model works reasonably well in simulating groundwater head regardless of the different parameter setting scenarios. However, the large difference of the absolute model-observation biases as represented by *Dmean*, *Dmedian*, *Dstd* suggest that the accuracy of our simulated groundwater head is sensitive to *RCH*, *K* and  $r_{rb}$ . The reference experiment where no adjustment on the two parameters is made shows the worst performance with three statistics of 452.76 m ( $D_{mean}$ ), 294.51 m ( $D_{med}$ ), and 494.59 m ( $D_{std}$ ), respectively (the lowest *R* as well, of 0.66). This means the simulated groundwater head is much shallower than the observations. This may be explained by the water balance at each grid cell: When K is low, the water exchange between adjacent cells is more difficult. With the amount of water entering each grid cell fixed (unchanged recharge) throughout the simulation, the slower water exchange between cells will result in more water accumulation within the cells and therefore higher water levels.

The simulated groundwater head is more sensitive to K compared to other parameters. For instance, in Table 1, when comparing experiments with identical values of RCH and  $r_{rb}$ , the simulation biases between experiments with different K values differ by several times, particularly when K is low. This is further seen in the spatial maps of model-observation bias in Figures C4. The simulated groundwater head is more sensitive to K in shallow groundwater areas (blue and green coloured areas, western U.S., Amazon, Sahel, the southern-north Eurasia, etc.) than in areas with deeper water tables (orange and red coloured areas, Rocky and Andes mountains, Tibetan Plateau, etc.). This pattern is consistent with the findings of de Graaf et al. (2015). However, our model's sensitivity to K is notably higher than that reported by de Graaf et al. (2015) and Reinecke et al. (2019 a,b). The Coefficient of Variation (CV) of the simulated heads exceeds 0.5 across most regions (not shown). We attribute it to three primary reasons: First, the number of our sensitivity analyses is limited. This may result in amplified standard deviation from individual extreme cases. Second, the model is poorly converged toward equilibrium under low K scenarios, especially in shallow groundwater occurrence regions. As illustrated in Figure 5 (the first column), groundwater heads in many of these areas exceed the drainage level, resulting in surface ponding. This forces us to tune K more favourably toward higher values in the sensitivity analyses, whereas the very low K scenarios stay unexplored. Third, compared to sensitivity analyses in Reinecke et al. (2019a), where only  $\pm 10\%$  perturbance on K is applied, our experiments feature a broader variability range of K.

The simulated head also shows sensitivity to groundwater recharge RCH and river bed conductance  $r_{rb}$ , but the sensitivity is more evident under low K scenarios. For example, the bias differences among the K0R0B0, K0R1B0, and K0R2B0 experiments are significantly larger than those observed in the corresponding experiments within Group A (e.g., K1R0B0,

K1R1B0, and K1R2B0) (Table 1). Moreover, in comparison to the corresponding experiments in Group B and Group C, the differences among K0R0B0, K0R0B1, and K0R0B2 biases even show orders of magnitude. These findings indicate that K remains the dominant hydrogeological parameter controlling groundwater head. At the same time, they also suggest that groundwater–surface water interactions – particularly the role of rivers – become crucial in regulating groundwater level fluctuations when lateral groundwater flow into or out of the aquifer system is limited due to low permeability. As a result, the simulation performance gradually improves as K increases; The improvement is further seen when  $r_{rb}$  decreases (which means more rapid river-aquifer exchange). To ensure further analyses are based on simulation with the highest realism, we chose the experiment with the best performance against observations as the baseline run (i.e., K2R1B2) for analyses in the following context. We also note that the model's performance could be further improved if more suitable combinations of the parameters were used. This can be achieved through observation-based bias correction procedures such as PEST (Doherty et al. 2003) and SCE-UA (Duan et al. 1992; 1993; 1994). However, since applying these algorithms globally is particularly time-consuming and the concentration of this study is to test the feasibility the established framework, the statistics from the current best-run experiment are reasonable enough for the time being, therefore we will leave further model improvement in future work.

### 3.2 Validation of Simulated WTD And the Sensitivity to Hydrogeological Parameters

The sensitivity of simulated WTD to the model's parameter settings does not follow the same way as the groundwater head (Table 2 and Figure 6 – Figure 7). Due to the small magnitude of WTD itself, an increasing of K yields only a marginal improvement in the median WTD bias  $(D_{med})$ , while the bias in standard deviation  $(D_{std})$  increases significantly (K0R0B0, K1R0B0, K2R0B0). The mean bias  $(D_{mean})$  shows a U-shaped response: It decreases initially, but once K exceeds a threshold, the bias grows again with opposite sign. The WTD response to Rch and  $r_{rb}$  is also less sensitive than the groundwater head, with only minimal improvement of  $D_{med}$  and  $D_{mean}$ . However, the response directions are within expectation. An increase of Rch yields shallower simulated WTD (e.g., K0R0B2 vs K0R1B2), whereas an increase of  $r_{rb}$  produces deeper simulated WTD (i.e., the bias shifts toward zero or positive) by enhancing drainage to channels (e.g., K1R0B1 vs K1R0B2).

A notable difference from the groundwater head is that the simulated WTD compares poorly to observations in all experiment runs at the global scale ( $R_{cor} < 0.3$ ) (Table 2 and Figure 6). The same poor behavior is also observed in the ensemble mean WTD from Reinecke et al. (2024) (Figure 7), suggesting this is a common problem in all global groundwater models. In addition to the model structure and parameter biases, we attribute this to several possible reasons below. First, since WTD is calculated as DEM minus groundwater head, it inherits bias from both inputs, which may result in exacerbated biases that can be of the same order as WTD itself; Second, there is a spatiotemporal mismatch between simulated and observed WTD. The Fan et al. (2013) dataset aggregates measurements from different years, with ~90% of locations having only a single reading; Moreover, each monitoring well in Fan et al. (2013) is a snapshot of local conditions. WTD can be highly heterogeneous within a 10 km × 10 km grid cell, so a single well may poorly represent the grid mean.

**Table 2.** Sensitivity experiment setting scenarios and the resulting WTD simulation statistics against observations. † Indicates the best-run experiment; ref for K indicates the mean hydrological conductivity from Gleeson et al. (2011); ref for Rch indicates the 40-year mean H08 recharge; ref for  $r_{rb}$  indicates 1 day. For the model and observation difference terms  $D_{mean}$  and  $D_{med}$ , positive values indicate the overall model head is deeper than observed head and vice versa.  $D_{std}$  is always positive; larger values indicate the overall simulated head deviates further from the observation.

|              | EXP                 | K          | Rch          | $r_{rb}$      | $R_{cor}$ | $D_{mean}$ | $D_{med}$ | $D_{std}$ |
|--------------|---------------------|------------|--------------|---------------|-----------|------------|-----------|-----------|
|              | K0R0B0              | ref        | ref          | ref           | 0.03      | -12.47     | -8.31     | 20.44     |
|              | K1R0B0              | $+1\sigma$ | ref          | ref           | 0.07      | -5.65      | -7.62     | 53.31     |
|              | K2R0B0              | $+2\sigma$ | ref          | ref           | 0.18      | 45.3       | -3.68     | 150.18    |
|              | K0R1B0              | ref        | $+0.5\sigma$ | ref           | 0.02      | -12.92     | -8.36     | 16.3      |
| A            | K1R1B0              | $+1\sigma$ | $+0.5\sigma$ | ref           | 0.04      | -9.49      | -8.02     | 39.05     |
|              | K2R1B0              | $+2\sigma$ | $+0.5\sigma$ | ref           | 0.14      | 24.49      | -5.09     | 117.97    |
|              | K0R2B0              | ref        | $-0.5\sigma$ | ref           | 0.1       | -2.74      | -7.5      | 63.87     |
|              | K1R2B0              | $+1\sigma$ | $-0.5\sigma$ | ref           | 0.17      | 21.7       | -5.92     | 120.87    |
|              | K2R2B0              | $+2\sigma$ | $-0.5\sigma$ | ref           | 0.22      | 115.49     | 2.67      | 258.59    |
|              | K0R0B1              | ref        | ref          | $\times 0.1$  | 0.04      | -11.85     | -8.23     | 23.81     |
|              | K1R0B1              | $+1\sigma$ | ref          | $\times 0.1$  | 0.1       | 0.29       | -6.71     | 66.88     |
|              | K2R0B1              | $+2\sigma$ | ref          | $\times 0.1$  | 0.19      | 61.27      | -0.67     | 163.61    |
|              | K0R1B1              | ref        | $+0.5\sigma$ | $\times 0.1$  | 0.03      | -12.56     | -8.3      | 18.61     |
| В            | K1R1B1              | $+1\sigma$ | $+0.5\sigma$ | $\times 0.1$  | 0.08      | -4.64      | -7.32     | 54.08     |
|              | K2R1B1              | $+2\sigma$ | $+0.5\sigma$ | $\times 0.1$  | 0.17      | 43.19      | -2.72     | 139.33    |
|              | K0R2B1              | ref        | $-0.5\sigma$ | $\times 0.1$  | 0.09      | -3.09      | -7.44     | 62.45     |
|              | K1R2B1              | $+1\sigma$ | $-0.5\sigma$ | $\times 0.1$  | 0.17      | 24.42      | -5.18     | 122.84    |
|              | K2R2B1              | $+2\sigma$ | $-0.5\sigma$ | ×0.1          | 0.22      | 106.1      | 13.76     | 228.28    |
|              | K0R0B2              | ref        | ref          | $\times 0.01$ | 0.04      | -11.66     | -8.13     | 24.44     |
|              | K1R0B2 <sup>†</sup> | $+1\sigma$ | ref          | $\times 0.01$ | 0.11      | 3.29       | -6.1      | 71.59     |
|              | K2R0B2              | $+2\sigma$ | ref          | $\times 0.01$ | 0.18      | 60.8       | 2.6       | 160.45    |
|              | K0R1B2              | ref        | $+0.5\sigma$ | $\times 0.01$ | 0.03      | -12.41     | -8.23     | 19.21     |
| $\mathbf{C}$ | K1R1B2              | $+1\sigma$ | $+0.5\sigma$ | $\times 0.01$ | 0.09      | -1.28      | -6.64     | 60.57     |
|              | K2R1B2              | $+2\sigma$ | $+0.5\sigma$ | $\times 0.01$ | 0.18      | 49.2       | -1.01     | 144.24    |
|              | K0R2B2              | ref        | $-0.5\sigma$ | $\times 0.01$ | 0.09      | -3.83      | -7.41     | 59.18     |
|              | K1R2B2              | $+1\sigma$ | $-0.5\sigma$ | $\times 0.01$ | 0.16      | 20.49      | -4.93     | 110.98    |
|              | K2R2B2              | $+2\sigma$ | $-0.5\sigma$ | ×0.01         | 0.19      | 79.78      | 10.57     | 186.63    |

**Figure 6**. Scatterplots of the simulated WTD under different parameter settings (Experiment group C). Inserted texts refer to statistics between model simulation (y-axis) and observation from Fan et al. (2013) (x-axis).  $R^2$ , n, Dmean, Dmed, and Dstd refers to coefficient of determination, sample size, mean, median and standard deviation of the simulation-observation difference.

Figure 7 Scatterplots of the simulated WTD against observations. WTD simulation in (a) is from H08-GM (experiment K2R1B2); WTD simulation in (b) is from ensemble mean in Reinecke et al. (2024).  $R^2$ , n, Dmean, Dmed, and Dstd refers to coefficient of determination, sample size, mean, median and standard deviation of the simulation-observation difference.

To investigate where the large WTD biases are presented, in Figure 8 we show the spatial maps as well as statistics of the model-observation biases of WTD from the best performance run over each continent. For North America where the highest observational density is presented, the model biases show a slightly left-skewed normal distribution. Approximately 3.9% of the analysed grid cells show biases within  $\pm 1$  m, 44.0% within  $\pm 10$  m, and 78.4% within  $\pm 50$  m. These grids are mostly located in the plain-dominated central and south-eastern U.S. The grid cells with large model-observation biases are distributed mostly over the mountainous areas but in a bimodal way. In the western U.S., the model tends to underestimate the groundwater head, whereas in the East the model tends to overestimate it. This can possibly be attributed to the uncertainty in aquifer properties, as well as the model's limitation in dealing with sharp groundwater head changes in mountainous areas. The topography in the western United States is comparatively higher, and the aquifer thickness is quite shallow (Figure 3(a)). The western mountainous areas mainly serve as the divergence region once it receives water from surface recharge. That is, the water will quickly move to adjacent lowlands due to the steep groundwater head gradient. The East, although also elevated, in fact serves as the convergence region due to the deeper aquifer thickness (Figure 3(a)). Over these areas of steep topographic gradient, the model simulation could become quite sensitive to the aquifer hydraulic conductivity setting. A large K scenario could possibly cause accelerated flow rate (therefore more water loss) in the West. On the contrary, a small K scenario would result in an overestimation of groundwater head in the West, as shown in Figure C5, where the bias is shown for the experiments glb K1R1B2, respectively. The polarity of biases is rather robust to K scenarios over other areas. Similar bias distribution is observed for other continents as well. For mountainous regions in the Alps and Brazilian Highlands, the model biases are quite pronounced; whereas for flatter areas such as the Netherlands and Northern Germany in Europe, Northern China Plain and Bangladesh in Asia, Amazon in South America, the model biases are minimal. Nonetheless, we note that the observations in Fan et al. (2013) inevitably embed the influence of human activity, whereas our model simulation is purely a natural run. The simulated groundwater level should be deeper than the current natural run if human water withdrawal were taken into account. This could lead to model-observation gap be skewed: Where the model head is higher than the observations (shallower WTD), the model-observation gap is exaggerated; where the model head is lower than the observations (deeper WTD), the gap is underestimated. The readers should bear this limitation in mind when interpreting the validation results.

Figure 8. Validation of simulated WTD against observations over each continent: (a) North America; (b) Europe; (c) Asia; (d) Africa; (e) Australia; and (f) South America. Grid cells are masked when either variable is marked as missing value. The missing values mainly

concentrate in western Australia (e), which results in a sharp edge in the centre of this region. The inset panels are histograms of the model—observation head residuals (h – ho) over each continent, with bar heights showing the count of sample pairs; the overlaid text annotations indicate the statistics of that residual distribution (mean, median, standard deviation, skewness).

To further investigate the climate and topography effects on WTD, we also show the WTD-slope relationship under waterlimited and energy-limited regimes respectively (Figure 9). The results show that for observations (left column in Figure 9). the correlation between slope and WTD is generally weak ( $\rho < 0.2, p < 0.001$ ) under both energy-limited and water-limited conditions. In contrast, the ensemble mean exhibits much stronger correlations (( $\rho \approx 0.6, p < 0.001$ ), suggesting that the multi-model mean tends to emphasize a stronger slope-WTD dependence. The Spearman  $\rho$  of H08-GM is closer to those of the observations; however, it should be noted that the relatively low numerical correlations may partly result from the large 460 variability of WTD within each slope bin. The much stronger correlation between medians and slope can still be observed. This pattern therefore suggests that current global numerical groundwater models may all tend to overemphasize the "groundwater head follows topography" relationship; Or in other words, they may potentially underrepresent the influence of other factors such as climate forcing and local aquifer properties. Additionally, we observe that in areas with smaller slope (e.g., below 10<sup>-3</sup> m m<sup>-1</sup>), the H08-GM simulated WTD (Figure 9, c and f) compares more closely to the observations (Figure 9, a and b). As the slope becomes steeper, the model-observation gap increases. The ensemble mean of Reinecke et al. (2024) 465 shows a similar pattern but a narrower spread within slope bins, likely reflecting two ensemble members that simulate systematically shallower WTD. The observed WTD is slightly deeper in water-limited regions than in energy-limited regions. The model captures this contrast, though the model-observation discrepancy is also modestly larger.

Since the flatter regions are often located with large cities and extensive human water use activities such as agriculture. We also evaluated the model performance of WTD in terms of cultivation and population density. Figure 10 shows that the simulated WTD compares reasonably well to observations in highly cultivated and populated areas. In regions with irrigation area fraction higher than 50% and population higher than 10,000/100km², both H08-GM and the ensemble-mean from Reinecke et al. (2024) compare closely to observations in terms of median and the 25th–75th percentiles. The ensemble-mean shows shallower WTD in regions with irrigation area fraction higher than 75%, while both H08-GM and ensemble-mean tend to overestimate WTD in highly populated areas. The wider interquartile range (25–75%) of WTD in H08-GM compared to the ensemble mean can be partly explained by the averaging nature of the ensemble. Since the ensemble mean combines the outputs from four different global groundwater models, two of which (Fan et al., 2013; Reinecke et al., 2019a) produce systematically shallower WTD (see Fig. 2 in Reinecke et al., 2024), the averaging process inherently smooths spatial variability and reduces the spread. On the other hand, the H08-GM simulations retain more of the spatial heterogeneity arising from its specific forcing data, lithological properties, and parameterization. Identifying which specific factors dominate this difference would require coordinated experiments under consistent simulation settings across all models.

However, we note that just the model's mean is similarly close to observations does not necessarily suggest the model's validity in specific places. In the stratified histograms provided in Figure 10 (b and c; e and f), the results show that even if highly irrigated and populated regions exhibit a narrower spread of WTD biases, the number of samples with small residuals (low bias) also increases across both low and high human-influence groups. Nevertheless, similar to the previous WTD-slope relationship, the analysis here reveals a systematic and interpretable relationship between human activities and the model's WTD biases at the global scale. A comprehensive site-level validation of model performance in terms of human gradients would be valuable in future.

Figure 9. WTD versus grid-scale slope binned on a logarithmic x-axis. Panels (a–c) show energy-limited regions; panels (d–f) show water-limited regions. Columns: (a,d) observations (blue), (b,e) ensemble products (green), and (c,f) H08-GM (orange). For each slope bin, boxplots summarize the WTD distribution (line = median; box =  $25^{th}$  and  $75^{th}$  interquartile range; whiskers indicate spread; gray dots, where shown, are individual samples).  $\rho$  indicates Spearman Correlation Coefficients between slope and WTD (\*\*\* indicates p 

Figure 10. Comparison of simulated and observed groundwater depth (WTD) under gradients of human activity. (a, d) Boxplots show the distributions of observed, ensemble-mean, and H08-GM WTD across bins of (a) irrigation fraction (0–25%, 25–50%, 50–75%, 75–100%) and (d) population density (people per 100 km²): 1–5K, 5–10K, 10–50K, 50–100K), respectively. Colors: observations from Fan et al. (2013) (blue), ensemble mean from Reinecke et al. (2024) (green), H08-GM (orange). For each bin, boxplots show the median (line) and 25<sup>th</sup> and 75<sup>th</sup> interquartile range (box); whiskers indicate spread. (b–c, e–f) Stratified histograms illustrate the relative frequency distribution of WTD residuals (simulated minus observed) for the H08-GM (b, e) and the multi-model ensemble (c, f), grouped by irrigation intensity (b–c) and population density (e–f).

### 3.3 Global Steady-state Groundwater WTD Maps

To investigate the spatial pattern of the simulated WTD from H08-GM, we illustrate the global WTD maps from all experiment runs in Figure 11 and Figure C6 – C7. Consistent to what has been observed in Section 3.2, the simulated WTD is more sensitive to hydrologic conductivity than the other two parameters, Rch and  $r_{rb}$ , e.g., the colour contrast from left to right of each row is much clearer than that from top to bottom of each column (Figure 12). The sensitivity seems to be higher in humid and flat regions, but this may be a visualization artifact influenced by the color-scale choice.

In Figure 12 we also present the global steady-state map of WTD from the best-run experiment from H08-GM (chosen as K2R0B2 by considering model-observation statistics of both groundwater head and WTD). The global WTD distribution shows a clear spatial gradient: the groundwater levels are considerably deep over the mountainous and arid regions whereas they remain shallow in flat and humid areas. The result corresponds well with previous studies as in Fan et al. (2013), de Graaf et al. (2015), and Reinecke et al. (2019a) and can be explained in the way that the mountains often serve as the divergence place for water to flow out due to their steep topography, and in arid regions the groundwater recharge from the surface is quite limited (vice versa). However, our result corresponds closer to the earlier works of de Graaf et al. (2015) and Reinecke et al. (2019a) than that of Fan et al. (2013) which is derived primarily from the observations in which the groundwater depth is up to 100m. Although partly applied the parameterization scheme of aquifer thickness (i.e., the *e*-folding factor) in Fan et al. (2013), the model framework in Reinecke et al. (2019a) largely follows MODFLOW. As such, the large gaps between the numerical and data-driven models here indicate careful comparison in model framework and parameterization schemes is needed to achieve cohesion in the two types of large-scale groundwater modelling studies.

Figure 11. Spatial maps of the simulated WTD to under different parameter settings (Experiment group C).

Figure 12. Global best-run steady-state Water Table Depth (WTD, meters below land surface).

### 3.4 Mechanisms Controlling Groundwater Distribution and Flow Dynamics

To help further understand the groundwater flow dynamics, in Figure 13 we present an analysis of the lower Mississippi River basin to showcase the complex interplays between groundwater flow and topography, aquifer and river hydrogeologic properties, and surface recharge. The high similarity between the spatial pattern of groundwater head and DEM (Figure 13, a and e), as well as the flow direction and velocity map (Figure 13, f), confirms the general principle that groundwater closely follows topography. However, the local characteristics in the north-western part of this region, where steep topography exists but limited groundwater flow present (shown as the low groundwater flow velocity and much deeper groundwater head compared to DEM), suggest aquifer properties that control the hydraulic gradient also play important roles in determining the water movement. The aquifer's *K* in these areas is much lower than in the other regions (Figure 13, b), which confirms this finding.

The role of surface recharge is only marginal in this case due to the strong heterogeneity of topography, but is evident in arid climate zones such as in Yellow River basin in Figure C8. The groundwater head distribution is jointly determined by both topography and recharge – in the northwest part of this region, although the topographic gradient is also sharp (Figure C9(a)), the recharge is quite limited (below 0.1 mm d<sup>-1</sup>) compared to the southeast high area. Consequently, the groundwater head over the low recharge area is consistently low and shows less spatial heterogeneity, regardless of the topographic gradient which plays an important role in the more humid climate regions.

River properties also play important roles in shaping the local characteristics of the groundwater head distribution through river-aquifer water exchanges. Although the groundwater head in Figure 13 appears much smoother than the topography map, we still observe the traces of major river channels, highlighting the significant role of river-aquifer exchange in determining the spatial distribution of the groundwater head. The topography pattern in Figure 13(a) aligns well with the river-aquifer water exchange rate pattern in Figure 13(h): Where there exists substantial water from groundwater to river (red colour), the groundwater head is lower than that in adjacent cells; Whereas where river supplies additional water to the aquifer (blue colour), the groundwater head is higher than the neighbour grid cells. The river-aquifer exchange rate is further determined by riverbed conductance and head difference between groundwater and river water channels. Most grid cells with higher water exchange rate (either positive or negative) tend to have higher riverbed conductance and larger river-groundwater head difference, which corresponds well to the governing equation in Eq. (1). Such river-aquifer exchange pattern is more evident in the Amazon River basins (Figure C9).

**Figure 13.** Spatial distribution of key parameters controlling groundwater flow and the resulting surface-groundwater interactions. The study region is in lower Mississippi River Basin. Panels (a–d) show model input variables, including the digital elevation model (DEM) (a), aquifer hydraulic conductivity (K) (b), riverbed conductance (c), and groundwater recharge rate (d). Panels (e–h) present simulated outputs, including groundwater head (e), lateral flow velocity with flow directions (f), head difference between aquifer and river (g), and river—

aquifer exchange rate (h), where positive values indicate losing rivers (water from rivers to aquifers) and negative values indicate gaining rivers (water from aquifers to rivers). Rectangular in white colour does not indicate missing values but extremely small values.

### 3.5 Global Net Groundwater Lateral Flows Estimated from H08-GM

As one motivation for developing the H08-GM model is to evaluate the compensating effect of groundwater lateral flow on urban water availability, in Figure 14 we also show the 41-year mean steady-state annual net lateral flow flux map. The net lateral flow flux here is calculated directly as flux convergence at each grid cell, and represents the net water fluxes a certain grid cell can gain or lose from the groundwater movement. The positive values indicate net inflow or groundwater "importers" (de Graaf and Stahl, 2022). Conversely, the negative values of the sum indicate net outflow or groundwater "exporters". The global pattern of groundwater lateral flow from the best-run simulation corresponds reasonably well with previous studies (Akhter et al., 2025; de Graaf and Stahl, 2022; Krakauer et al., 2014; Miguez-Macho and Fan, 2025). The highest net lateral flow distributed in Amazon, highlighting its critical role in sustaining the ecosystem in its neighbourhood. Moderately high flows are observed in the eastern United States, Central Africa, north-western Eurasia, and the tropical islands. Amazon serves as the world's largest groundwater exporters.

In terms of magnitude, our results compare more closely with those of (de Graaf and Stahl, 2022), reaching over 600 mm yr<sup>-1</sup> in high flow regions. However, this is considerably higher than the 100 mm yr<sup>-1</sup> reported by (Krakauer et al., 2014), while much lower than the 1000 mm yr<sup>-1</sup> maximum estimated in (Miguez-Macho and Fan, 2025). Two possible reasons may explain this discrepancy. The first is the scale-dependence of lateral flow flux. Previous studies have shown that simulated groundwater lateral flow flux tends to increase as the spatial resolution of a model becomes finer (Akhter et al., 2025; Krakauer et al., 2014). The results of (Miguez-Macho and Fan, 2025), estimated at a 1 km resolution, therefore represent a finer-scale simulation that naturally yields higher flow magnitudes. Second, the estimated flow flux is strongly influenced by the model's paramter settings, especially the hydraulic condcutivity. When relatively small hydraulic conductivity values are used, the flux magnitude decreases significantly (Figure C10, left column). However, even under the lowest hydraulic conductivity scenario, the ratio of net lateral flow flux to groundwater recharge can still be high, suggesting the lateral groundwater flow plays a nonnegligible role in the grid cell's water budget.

**Figure 14.** Global distribution of simulated net lateral groundwater flow (mm yr<sup>-1</sup>) derived from the coupled H08-<u>GM</u> model. Overlaid on the map are major global cities categorized into water-scarce (orange inverted triangles) and non-water-scarce (black upward triangles) groups (Mekonnen and Hoekstra, 2016). Positive values indicate net groundwater flow "exporters" and negative values indicate "importers". See Figure C11 for a Zoomed-in version.

The net lateral flow results highlight the important role of the compensating effects of groundwater flows in sustaining regional water budgets, which should be considered but have long been downplayed in GWMs. In Figure 14 we also overlaid several megacities in the world, classified as water-scarce and non-water-scarce categories based on Mekonnen and Hoekstra (2016). It is clearly observed that the groundwater lateral flow effect, whether it be importers or exporters, is quite considerable in some water-scarce cities, e.g., Beijing, Houston, etc., with net groundwater flow higher than 100 mm yr<sup>-1</sup>. For other non-water-scarce cities as Tokyo, Berlin, New York, etc., the net groundwater flow is even higher, approaching 200 mm yr<sup>-1</sup>. The large amount of net groundwater flow must be explicitly incorporated into current water resource management models: Neglecting "exporters" effect may underestimate the city's water stress while neglecting "importers" effect may tend to overestimate it. However, we note that this analysis is only an exploratory illustration to show that our model has potential for the representation of megacities in GWMs. The operational city-scale groundwater lateral inflow/outflow assessments require more robust analyses to address the model's spatial resolution, the uncertainties in aquifer hydraulic conductivity, riverbed conductance, and other boundary conditions.

### **4 Conclusions**

This study has presented a high-resolution global groundwater model H08-GM by incorporating various global hydrogeological datasets. Sensitivity analyses have been conducted on several key model parameters in order to produce the best model performance in simulating the steady-state groundwater levels. Validated against approximately 1.6 million in-situ observations, the results show that the model with optimal parameter settings performs well at the global scale with *R* of 0.93. The model performs particularly well over plain areas where large cities and extensive human activities are located, with groundwater head biases within ±25m, but the model tends to show larger biases over mountainous regions, possibly due to the uncertainty in aquifer properties as well as model's limitation in dealing with sharp groundwater head changes. Our results demonstrate that the coupled H08-GM modelling framework can effectively reproduce realistic spatial gradients of groundwater heads, with deeper groundwater tables in mountainous areas from shallower groundwater in plains. Such a pattern primarily results from the topographically driven groundwater flow dynamics, with aquifer and river hydrogeological properties contributing significantly to the local heterogeneity. Using the model, we identify the regions that function as net groundwater "importers" or "exporters" at the global scale and show that the annual net groundwater lateral flow amount can be quite considerable, in the magnitude nonnegligible to annual surface groundwater recharge. This highlights the important role of the groundwater lateral flow in maintaining regional water budget and has to be considered in water resources models, particularly for megacities.

Several limitations should be noted for potential model users. First, our model only applies a single unconfined aquifer layer, and thus omits vertical head gradients, aquitard leakage, and coastal effects that are central to deep confined basins (e.g., Northern China Plain, Central Valley, etc.). Consequently, the simulated groundwater head over the areas with deep confined aquifer system can be underestimated. This simpler model conceptualization was chosen due to the limited availability of global confined aquifer hydrogeological parameters and the evidence that the shallow groundwater (mainly unconfined) contributes largely to sustain anthropogenic and ecological groundwater use purposes (Gao et al., 2018). Second, the current simulation is still one-way (H08  $\rightarrow$  MODFLOW) with no feedback from groundwater to land-surface processes. As a result, the excessive groundwater is simply removed from the aquifer system, rather than enters to the surface water to strengthen their recharge to the aquifer. H08 evapotranspiration and allocation do not respond to the simulated groundwater heads or capillary rise; river water level is not updated by modeled baseflow. This could cause underestimation (deeper) of the simulated groundwater head than it should be if the two-way simulation were enabled. Furthermore, there still exist uncertainties in the model's key hydrogeological parameters. Compared with the earlier global sensitivity analysis by de Graaf et al. (2015), which mainly evaluated the coefficient of variation of model outputs, and the more comprehensive subsequent study by Reinecke et al. (2019b), which systematically quantified model sensitivity to both individual and combined parameter variations through an extensive set of 1,848 Monte Carlo experiments, our OAT sensitivity test provides a complementary but more limited perspective on parameter uncertainty. However, the fact that the simulated groundwater heads compare reasonably well to the

in-situ observations globally confirms the feasibility of our model, although more comprehensive parameter tunings are suggested in the future.

Our model contributes as one of the three major GHWMs that explicitly considers groundwater lateral flow at the global scale.

Additionally, the capability of H08-GM to directly output groundwater levels, calculate lateral flow rate, and connect rivers and aquifers, provides a powerful tool to investigate the groundwater decline trend over the pumping hotspots in the world, to identify river basins as importers or exporters, and to examine the losing and gaining regimes of streamflow. It will essentially help improve the accuracy of the water resource availability estimated based on the original H08 model. The steady-state simulation result in this paper has demonstrated the 40-year mean natural groundwater level distribution without human disturbance. In the next step, the temporal groundwater level variability and the human water withdrawal effect over the past 40 years should be investigated to help further advance our understanding of the important role of groundwater in supporting human water consumption, and the fundamental mechanisms behind the human-groundwater interactions.

### Appendix A: Algorithms to calculate river channel depth and river width

In the latest version of CaMaFlood, the river channel depth  $(D_{chn})$  is calculated based on the power-law empirical equation, as:

$$D_{chn} = max (H_{min}, H_c * Q_{chn}^{H_p} + H_0)$$
 (A1)

Where,  $H_{min}$ = 1.0 is the prescribed minimum channel depth (unit: m);  $H_c = 0.1$  and  $H_p = 0.50$  are the coefficients,  $H_0 = 0.00$  is the prescribed offset number for river channels;  $Q_{chn}$  is the river discharge (unit: m<sup>3</sup> s<sup>-1</sup>).

The river width  $(RIV_{wth})$  is obtained based on both satellite observation and power-law estimation. The satellite-derived river width is first read in as the baseline variable  $(RIV_{gwdlr})$ . The river width based on power-law  $(RIV_{wth})$  is then calculated separately, as:

$$RIV_{wth} = max \left( W_{min}, W_c * Q_{chn}^{W_p} + W_0 \right) \tag{A2}$$

Where,  $W_{min} = 5.0$  is the prescribed minimum river channel width (unit: m),  $W_c = 2.50$  and  $W_p = 0.60$  are the coefficients, and  $W_0 = 0.00$  is the prescribed offset number;  $Q_{chn}$  is the river discharge (unit: m<sup>3</sup>/s).

Afterwards,  $RIV_{wth}$  is used to constrain the underestimation of  $RIV_{gwdlr}$  for small rivers and overestimation for large rivers, as:

$$RIV_{gwdlr} = \begin{cases} \max(RIV_{gwdlr}, RIV_{wth}), & if \ RIV_{gwdlr} < 50 \\ RIV_{wth} * 0.5, & if \ RIV_{gwdlr} < RIV_{wth} * 0.5 \\ RIV_{wth} * 5.0, & if \ RIV_{gwdlr} > RIV_{wth} * 5.0 \\ 10000, & if \ RIV_{gwdlr} > 10000 \end{cases} \tag{A3}$$

# Appendix B. Look-up table for lithology-based aquifer conductivity (log-transformed, m²).

sample size, and standard deviation  $(\sigma)$  for each subcategory are listed. The category-averaged mean  $(\overline{\mu_{log K}})$  and standard deviation  $(\overline{\sigma_{log K}})$  are then (Unconsolidated Sediments), SS (Siliciclastic Sedimentary Rocks), ND (No Data), PB (Basic Plutonic Rocks), SM (Mixed Sedimentary Rocks), The 16 lithology categories are PY (Pycroclastics), VB (Basic Volcanic Rocks), PA (Acid Plutonic Rocks), MT (Metamorphic Rocks), SU WB (Water Bodies), VI (Intermediate Volcanic Rocks), SC (Carbonate Sedimentary Rocks), VA (Acid Vocanic Rocks), EV (Evaporites), PI (Intermediate Plutonic Rocks), IG (Ice and Glaciers). For each lithological categories there are maximum 4 subcategories (SbC). The mean ( $\mu$ ), calculated and given in the last two columns.

| G (m <sup>2</sup> )            | Logk (III)  | 2.0   | 1.8   | 1.5   | 1.5   | 1.8   | 2.1   | 0.0     | 1.5   | 2.1   | 0.0   | 1.8   |
|--------------------------------|-------------|-------|-------|-------|-------|-------|-------|---------|-------|-------|-------|-------|
| (m <sup>2</sup> )              | PlogK (***) | -13.2 | -12.5 | -14.1 | -14.1 | -12.6 | -15.0 | -20.0   | -14.1 | -15.0 | -20.0 | -12.5 |
|                                | SbC4        | 0.0   | 0.0   | 0.0   | 0.0   | 2.0   | 2.5   | N/A     | 0.0   | 2.5   | N/A   | 0.0   |
| (m <sup>2</sup> )              | SbC3        | 1.8   | 0.0   | 0.0   | 0.0   | 1.8   | 1.7   | N/A     | 0.0   | 1.7   | N/A   | 0.0   |
| $\sigma_{logK}~(\mathrm{m}^2)$ | SbC2        | 2.5   | 1.8   | 1.5   | 1.5   | 1.2   | 6.0   | N/A     | 1.5   | 6.0   | N/A   | 0.0   |
|                                | SbC1        | 2.0   | 1.8   | 1.5   | 1.5   | 2.0   | 2.5   | N/A     | 1.5   | 2.5   | N/A   | 1.8   |
|                                | SbC4        | 0     | 0     | 0     | 0     | 113   | 20    | N/A     | 0     | 20    | N/A   | 0     |
| Sample size (-)                | SbC3        | 33    | 0     | 0     | 0     | 31    | Ξ     | N/A     | 0     | Ξ     | N/A   | 0     |
| Sample                         | SbC2        | 20    | 33    | 17    | 17    | 82    | 6     | N/A     | 17    | 6     | N/A   | 0     |
|                                | SbC1        | 113   | 33    | 17    | 17    | 113   | 20    | N/A     | 17    | 20    | N/A   | 33    |
|                                | SbC4        | 0.0   | 0.0   | 0.0   | 0.0   | -13.0 | -15.2 | N/A     | 0.0   | -15.2 | N/A   | 0.0   |
| (m <sup>2</sup> )              | SbC3        | -12.5 | 0.0   | 0.0   | 0.0   | -14.0 | -16.5 | N/A     | 0.0   | -16.5 | N/A   | 0.0   |
| µюдк                           | SbC2        | -15.2 | -12.5 | -14.1 | -14.1 | -10.9 | -12.5 | N/A     | -14.1 | -12.5 | N/A   | 0.0   |
|                                | SbC1        | -13.0 | -12.5 | -14.1 | -14.1 | -13.0 | -15.2 | N/A     | -14.1 | -15.2 | N/A   | -12.5 |
| Cat                            |             | ΡΥ    | VB    | PA    | MT    | SU    | SS    | N<br>ON | PB    | SM    | WB    | VI    |

continued)

| 1.5         | 1.8   | 0.0   | 1.5   | (    |
|-------------|-------|-------|-------|------|
| -11.8       | -12.5 | -20.0 | -14.1 |      |
| 0.0         | 0.0   | N/A   | 0.0   | 11/1 |
| 0.0         | 0.0   | N/A   | 0.0   | A1/A |
| 0 1.5 1.5   | 1.8   | N/A   | 0.0   | A1/A |
| 1.5         | 1.8   | N/A   | 1.5   | A1/A |
|             | 0     | N/A   | 0     | A1/A |
| 0           | 0     | N/A   | 0     | A1/A |
| 47          | 33    | N/A   | 0     | NI/A |
| 47          | 33    | N/A   | 17    | N1/A |
| 0.0         | 0.0   | N/A   | 0.0   | A1/A |
| 0.0         | 0.0   | N/A   | 0.0   | N1/A |
| -11.8 -11.8 | -12.5 | N/A   | 0.0   | ATTA |
| -11.8       | -12.5 | N/A   | -14.1 | NI/A |
| SC          | VA    | EV    | PI    | 7    |

# **Appendix C: Supplementary figures**

**Figure C1.** Global distribution of lithology category. The 16 lithology categories are PY (Pycroclastics), VB (Basic Volcanic Rocks), PA (Acid Plutonic Rocks), MT (Metamorphic Rocks), SU (Unconsolidated Sediments), SS (Siliciclastic Sedimentary Rocks), ND (No Data), PB (Basic Plutonic Rocks), SM (Mixed Sedimentary Rocks), WB (Water Bodies), VI (Intermediate Volcanic Rocks), SC (Carbonate Sedimentary Rocks), VA (Acid Vocanic Rocks), EV (Evaporites), PI (Intermediate Plutonic Rocks), IG (Ice and Glaciers).

**Figure C2** Global distribution of groundwater recharge statistics for sensitivity analyses. (a) Same as Figure 2(a) in the main context: 41-year average groundwater recharge rate (mm d<sup>-1</sup>); (b) Standard deviation of groundwater recharge rate (mm d<sup>-1</sup>); (c) Groundwater recharge of 41-year mean plus 0.5 standard deviation (mm d<sup>-1</sup>); and (d) Groundwater recharge of 40-year mean minus 0.5 standard deviation (mm d<sup>-1</sup>).

Figure C3 Global distribution of water-limited and energy-limited regions.

Figure C4 Validation of simulated groundwater head against observations over each continent: (a) North America; (b) Europe; (c) Asia; (d) Africa; (e) Australia; and (f) South America. The observed groundwater head is obtained as surface elevation minus WTD, both of which are directly from the report in Fan et al. (2013). Grid cells are masked when either variable is marked as missing value. The missing values mainly concentrate in western Australia (e), which results in a sharp edge in the centre of this region. The inset panels are histograms of the model–observation head residuals (h – ho) over each continent, with bar heights showing the count of sample pairs; the overlaid text annotations indicate the statistics of that residual distribution (mean, median, standard deviation, skewness).

Figure C5 Simulated groundwater head bias from experiment glb K1R1B2

Figure C6. Global WTD distribution from sensitivity experiments Group A

Figure C7 Global WTD distribution from sensitivity experiments Group B

Figure C8 Groundwater flow dynamics for Yellow River basin

Figure C9 Groundwater flow dynamics in Amazon River basin

**Figure C10** Net lateral flow flux (left column) and ratio of net lateral flow flux to annual groundwater recharge (right column) under different hydraulic conductivity scenarios. K0, K1, and K2 indicates the original hydraulic conductivity, hydraulic conductivity adjusted by one standard deviation, and hydraulic conductivity adjusted by two standard deviations, respectively. Note the colorbar range in the left column is different.

Figure C11. Zoomed-in version of the simulated net lateral groundwater flow (mm yr<sup>-1</sup>) derived from the coupled H08-GM model.

# Code and Data Availability

- H08-GM v1.0 is open source and distributed under the terms of Creative Commons Attribution 4.0 International License. The development model tools and all data input of H08-GM are available in a Zenodo repository (doi: 10.5281/zenodo.15709184). The development and maintenance of H08-GM are conducted at the Department of Civil Engineering, The University of Tokyo. We welcome researchers from external institutes to contribute.
- He, Q., Hanasaki, N., Matsumura, A., Sutanudjaja, E., & Oki, T. (2025). Release of H08-GM(v1.0) code (steady-state). Zenodo. https://doi.org/10.5281/zenodo.15709184

## Supplement.

All supplementary materials can be found in Appendix attached in this manuscript.

#### **Author Contributions**

QH and NH conceptualized this work. QH performed methodology, implementation for all workflows, pre-processing of hydrogeological data, the simulation of MODFLOW 6, and analyses of the simulation results. AM prepared the global H08 model output. EHS helped with preparation of the aquifer thickness data and several technical issues of MODFLOW simulation. NH and TO supervised this research. QH prepared the manuscript, with contributions from all authors.

#### 745 Competing Interests.

The contact author has declared that none of the authors has any competing interests.

#### Disclaimer.

#### 750 Acknowledgments

We thank Prof. Ying Fan for providing the global steady-state groundwater head observation data. QH appreciates valuable discussions with Prof. dr. ir. Marc F. P. Bierkens, Dr. ir. Inge de Graaf, Dr. Sida Liu, and other group members during the stay in Utrecht University, as well as the discussions with Prof. Reed Maxwell, Prof. Chunmiao Zheng, and other scientists during the GEWEX groundwater workshop. The authors would like to thank Dr. Robert Reinecke and two anonymous reviewers for their constructive comments which greatly help improve this manuscript.

# Financial Support.

This work was partially supported by Japan Society for the Promotion of Science (KAKENHI; 21H05002) and the

760 Environment Research and Technology Development Fund (JPMEERF23S21120) of the Environmental Restoration and

Conservation Agency, Ministry of the Environment of Japan. QH appreciates financial support from JSPS International

Research Fellow program (ID No. 24096).

# 765 Review Statement

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
