# Peer review of "Development of A Global 5arcmin Groundwater Model (H08-GMv1.0): Model Setup and Steady-State Simulation"

_EGUsphere, 2025_

## Author Comment (AC1)

**Dear Dr. Cenlin He,**

Thank you so much for handling our manuscript. Below, we respond to the three reviewers' feedback as well as comments from the community. We have improved the manuscript in several aspects:

- We have substantially revised the manuscript context to enhance the overall readability, and refined all color maps using the Scientific Colour Map 7.0 recommended by GMD;
- We have included the validation of groundwater table depth (WTD) to complement the previous evaluation based solely on groundwater head;
- We have further assessed the model performance across different climate regions and human activity gradients, to better understand the functional relationships underlying groundwater dynamics.
- We have added comparisons with ensemble mean results from other global groundwater models;
- And finally, we have clarified several technical details of the model implementation and elaborated the discussion of model limitations.

By incorporating these improvements, we hope that we have satisfactorily addressed all reviewer comments and that the revised manuscript will meet the standards for publication in GMD

Below, we provide our responses in blue text, while the reviewers' original comments remain as black.

Sincerely regards,
Qing He
on behalf of all coauthors

**Reviewer #1, Dr. Robert Reinecke**

In their manuscript, He et al. present a steady-state groundwater model forced by outputs of the global hydrological model H08.

In general, this study is very timely, and it is nice that H08 is also approaching this difficult task of including a gradient-based groundwater component. However, there are also several areas where this manuscript needs improvement.

- 1) As already mentioned in an early community comment, some studies provide some more context for the finding presented here.
- 2) Instead of the global maps, a direct comparison to existing published results would significantly improve the scientific value of this study. I made a concrete suggestion further below. Parts section 3.5 adds not much and should either be extended or removed.
- 3) The sensitivity analysis is currently a manual calibration and not a classical sensitivity analysis. This needs either to be named manual calibration or the one traditional goal of sensitivity analysis needs to be achieved (see also more details below.
- 3) And finally, I suggest also reconsidering the framing of the study because what the authors present is not yet a coupled model but rather a global steady-state groundwater model forced by aggregated inputs from H08. Either the authors need to add at least ideas on how the complete two-way coupling can be implemented, or they need to adapt how the results are presented.

With regards,

Robert Reinecke

**Response:**

Dear Dr. Reinecke, we sincerely appreciate your constructive and insightful comments. We have carefully addressed all the points raised.

In particular, regarding your last comment, we agree that this paper only shows the steady-state groundwater simulation, which is stated in both the title and throughout the manuscript. To emphasize that the two-way coupling will be an indispensable part of our model development work in the future, we have added a discussion in Section 2.1 outlining our ongoing efforts and conceptual plans for the two-way dynamic coupling H08-MODFLOW simulation.

**Detailed comments:**

16: but then it is a manual calibration, not a sensitivity analysis

Response: Thank you for raising this point. We have revised the terminology to "Local One-At-A-Time (OAT) Sensitivity Tests" rather than "manual calibration". See our detailed response to another comment below (Line 15).

18: Two previous studies? There is much more than that. Also, it is unclear what this refers to here. Studies on H08 or more generally global groundwater modeling Response: Thank you. We have revised the wording here (Line 16-20).

17: Does this refer to WTD or head? This needs to be clarified because I suspect it is head.

Response: It is head. We have clarified accordingly (Line 16-20).

**20: Did you mean shallow?**

Response: We apologize for the typo. The original sentence has been removed to better fit the new abstract context.

35: Groundwater is also in itself an ecosystem Sacco et al (2023).

Response: Thanks. Citation added (Line 31).

40: You don't really model salinity here. I suggest removing this. Even if intrusion and discharge are critical processes, this deviates too much from the story you want to tell here

Response: Thanks. We have removed the sentence.

51 and following: I don't know if this focus on these specific models is necessary. An increasing number of models that represent the terrestrial global water cycle are starting to include groundwater as an explicit component. There is a plethora of models that we could call global water models

(https://wires.onlinelibrary.wiley.com/doi/full/10.1002/wat2.70025). Especially since you then also discuss Parflow. A more general framing of the idea that there is an interest in better representing the global water cycle is better than trying to categorise H08 with two other models as the only global hydrological models out there, which is also not true.

Response: Thank you so much for this constructive suggestion. We have rephrased the context here and hope it is now of a broader interest to the global water model community (Line 46-58).

**82: Reinecke instead of Reneicke**

Response: We are terribly sorry for the typo. It is corrected (Line 66).

93: Does this refer to a 41-year mean of conditions that were used to force a steady-state model where time was removed from the model formulation to reach an equilibrium, or did you use a transient formulation to reach a defined steady-state? This is unclear here. Also, why this period and not, e.g., 1901, which is often the starting point for ISIMIP simulations?

Response: The steady-state in our study refers to the previous one, i.e., time was removed from the model formulation. We have clarified this in the revised manuscript. Also, we chose 1979 - 2010 as the simulation period just to align with an ongoing high-resolution H08 simulation project that uses the same time span. Because the main purpose of this study is model development, the chosen period is meant as a test case. For further analyses and integration into ISIMIP, we will no

doubt extend the simulation period to align with the protocol. We have clarified the definition of the steady-state in the revised manuscript (Line 78-79).

Also, the rest of the sentence is unclear - included in what?

Response: We meant "included in this study". Corrected in the revised manuscript (Line 81).

112: This is a limitation that should be mentioned in the abstract.

Response: Thanks. We have revised the abstract accordingly (Line 25).

175: Why this exact period and not, e.g., 1901 or a mean of 100 years? Response: We did it just to test our model performance and to align with one another's H08 simulation study. See our previous response.

176? Why are the aggregated to monthly when you are running at monthly time steps (line 175)? Or does that mean you calculated the arithmetic mean over this period? Ah, it is described in the following sentence. Please be sure to improve the clarity in your manuscript.

Response: Thank you. We appreciate your careful reading.

203: This is not correct. It has not been proven in any scientific way other than that it provides decent results, however, not better than in models that don't use this approach. Essentially, it is a calibration of transmissivity to observed heads. I am not saying it is wrong, but it should be introduced correctly.

Response: Thank you for pointing this out. We have revised the wording (Line 196).

Fig. 3a Please use appropriate colors - also in other figures. https://hess.copernicus.org/articles/25/4549/2021/hess-25-4549-2021.html is a good read concerning this. Also, the figure text does not include the units shown.

Response: We apologize for the inappropriate choice the colormap. We have now applied the colormaps from Scientific colour maps 7.0 to facilitate a more CVD-friendly visualization of our results for this figure and throughout the manuscript.

**218: Is there a reason why you chose to not use the permafrost categorization to limit conductivity in the North?**

**Response:**

Thank you for the suggestion. We have re-run all the simulations by constraining permafrost hydrological conductivity based on permafrost zone index (pzi) map from Gruber 2012 (see the spatial extent below), similar to Reinecke et al. 2019, HESS. However, constraints on hydraulic conductivity are only applied to regions with pzi > 0.5, as this threshold corresponds to continuous and extensive permafrost zones where temperature fluctuations and freeze—thaw dynamics strongly influence K values.

Different from Reinecke et al. (2019, HESS), we did not assign a uniform hydraulic conductivity (K) value of 10-13 m/s to all permafrost grid cells. Instead, we applied a scale factor to the original K when the soil is unfrozen. This approach was adopted for two main reasons:

- (1) The model exhibited poor convergence performance when using an extremely small K, with many grid cells showing simulated groundwater heads higher than the land surface; and
- (2) The change in K within permafrost regions is controlled by multiple factors,

including soil temperature, freeze—thaw dynamics, and soil texture. Previous studies have shown that the K can decrease by one to four orders of magnitude when transitioning from unfrozen to frozen conditions (Watanabe and Flury, 2008, WRR, doi: 10.1029/2008WR007012; Watanabe and Osada, 2016, VZJ, doi: 10.2136/vzj2015.11.0154).

To maintain numerical stability in our simulations, we adopted a conservative estimate, representing a one-order-of-magnitude reduction in K. We admit this is a strong assumption, and we have explicitly acknowledged this in the revised manuscript (Line 214-217).

230: Which is consistent with https://hess.copernicus.org/articles/23/4561/2019/ What does "1d" refer to?

Response: Thank you for recommending this paper. The river—aquifer exchange scheme in Reinecke et al. (2019) adopts the standard head-dependent conductance formulation, which is also used in MODFLOW. Our implementation follows the same formulation. Given this equivalence, we chose not to particularly highlight the scheme consistency here to avoid redundancy, while cite Reinecke et al. (2019) in the other part of this manuscript.

"1d" means "1 day". Corrected (Line 224).

Figure 4: Bad figure quality. The subfigure references have different sizes, and sometimes they are in the figure. C is missing a legend altogether.

Response: Thanks. We have adjusted them accordingly.

250: The DEM is at a higher resolution than 5-arcmin. How did you determine the river elevation exactly?

Response: The spatial resolution of DEM is aggregated to 5-arcmin using linear interpolation algorithm to ensure consistency with other dataset, similar to what has been done in Reinecke et al. (2024). The assumption is that the influence of the simplified upscaling method would only have effect on sub-grid topographic variability and only have negligible impact on the first-order global pattern of simulated head. We have clarified the data processing technique in the reivesed manuscript (Line 250 - 251).

280: While it is great that the authors did this. It is also important to frame it correctly. This is a local one-at-a-time SA which does not account for interactions. And since you don't achieve any of the traditional goals of SA, i.e., screening, ranking, or mapping https://www.sciencedirect.com/science/article/pii/S1364815216300287, I would say it is rather a manual calibration than a SA.

Response: We appreciate the reviewer's clarification. We agree the analyses here do not rigorously follow the traditional SA standard. However, we are hesitant to name this part as manual calibration since we did not define or minimize any objective function, nor perform iterative tuning. We'd rather prefer calling it "exploratory local (OAT) sensitivity tests" as the reviewer suggested, and explicitly state the limitation of our analyses.

282: see also Reinecke et al. (2021) for recharge, (2019) for other parameters, (2020) for spatial resolution, and (2024) regarding simulated WTD Response: Thank you so much for the recommendation of the papers and the great contribution you've done. We have added relevant discussion in the revised manuscript (Line 283, Line 291, Sections 3.1 and 3.2).

285: RRB is upper case in the text and r\_rb in the table. Please be consistent. Also, what does "ref" mean? What is the baseline value for this?

Response: Thanks for the careful reading. ref for K indicates the mean hydrological conductivity from Gleeson et al. (2011); ref for Rch indicates the 40-year mean H08 recharge; ref for r\_rb. indicates 1 day. We have made the wording consistent and added explanation in Table 1's caption.

304: I would disagree with this. The head is what the model actually simulates, but what is relevant for many applications, e.g., in determining whether groundwater is available to humans or ecosystems, is WTD. Furthermore, calculating statistical metrics such as an error metric on the head is likely very biased by the topographic influence that is encoded in the head distribution. Scatterplots look very different depending on whether head or WTD is shown. Please show 1) scatterplots of simulated head vs. observed and scatterplots of simulated vs. observed WTD. See

also Reinecke (2020) for a related discussion. Furthermore, I suggest comparing your outputs to other existing steady-state simulations. You could even use the ensemble published with Reinecke et al. (2024).

**Response:**

We thank the reviewer for highlighting this important point. We agree that water-table depth (WTD) is the more application-relevant metric (e.g., for human/ecosystem groundwater accessibility). We have therefore prepared the scatterplots of WTD between H08-GM and Observation, and compared it with the results with the ensemble-mean WTD from Reinecke et al. (2024) as well (Section 3.2 in the revised manuscript).

As expected, the model-simulated WTD, no matter from H08-GM or ensemble mean, compares poorly to the in-situ observations, suggesting this might be a common problem in all global groundwater models. Two reasons could be plausible: (1) WTD embeds biases from both DEM and groundwater head so that the bias of itself might be exacerbated; (2) The spatiotemporal inconsistency of WTD in Fan et al. (2013) and the model simulations, i.e., most observations of Fan et al. (2013) only have one reading, and since they are in-situ observations, they cannot represent the 10km x 10km footprint. See a more detailed discussion in the revised manuscript (Line 390 – Line 398).

As the reviewer suggested in a later comment, we have also compared the WTD-slope functional relationships under different climate regimes in the figure below (top row for the energy-limited regime and bottom for the water-limited). H08-GM resembles the ensemble-mean in both water-limited and energy-limited regions, but neither of the models' results follow closely to the observations.

However, we believe the model can perform reasonably well in some regions, if not in the global sense. Inspired by Reinecke et al. (2024), we evaluated the performance of H08-GM and ensemble-mean in terms of different irrigation area fractions and population densities. The results suggest that in high irrigation fraction areas (e.g., >50%) and high population density areas (>10K/100km2), both H08-GM and ensemble-mean show closer median and model spread of WTD compared to the observations (Figure next page). This is important since it is in these areas the groundwater matters more to human and agricultural water accessibility.

In conclusion, we agree that the poor model-obs WTD problem in H08-GM, so as in other global groundwater models, but the models can perform reasonably well in densely populated and irrigated regions. This should be an important research

direction in the future for the global groundwater modeling community. To highlight topic, we have included the above content in the newly added Section 3.2 in the revised manuscript.

310: Again, this is an artifact of using head instead of WTD and suggests a much better model performance than actually is the case. Since the model is likely performing very well in shallow aquifers but much worse in deeper aquifers. Response: We have revised the sentence to underscore this is for groundwater head (Line 335).

370: What is the impact of comparing human-impacted observation to simulation based only on a natural run? What deviation can be explained by this, and which are the model limitations?

**Response:**

This is an important point and we appreciate the reviewer's scrutinization. The observations in Fan et al. (2013) inevitably embed the influence of human activity, whereas our model simulation is purely a natural run. The simulated groundwater level should be deeper than the current natural run if human water withdrawal were taken into account (depends on region). This could lead to the model-observation gap being skewed: Where the model head is higher than the observations (shallower WTD), the model-observation gap is exaggerated; where the model head is lower than the observations (deeper WTD), the gap is underestimated. We have included these limitations in the revised manuscript (Line 435 - 440).

Fig. 5, 7: Please also adjust the color here.

Response: Thanks. Adjusted.

Fig.5: The small maps are not very helpful. How about showing the results of the manual calibration in terms of different error metrics (bias, root-mean squared, max deviation) and scatterplots of head and WTD here. This would be much more informative.

Response: Thank you for the suggestion. We believe that in addition to the scatterplots and error metrics, the maps here are important to show the spatial distribution of the simulated WTD. We have now included the scatterplots in Section 3.1 and 3.2, and moved the maps to the Section 3.3, which is specifically to show the spatial distribution of WTD.

Fig. 7: Instead of showing a direct comparison, e.g., as a difference map or scatter plots, to the existing results would be more informative than a global map. Consider comparing the functional relationships to the slope we propose in 2024 as well. Response: Thanks. We have investigated this interesting relationship and show the results in Figure 9 in the revised manuscript. See our response to your previous comment.

Fig.8: Log scale of the river bed conductance? And where exactly is this zoom in from?

Response: Thank you. We have adjusted the map to show log-scale river bed conductance. The location is in lower Mississippi river basins, and we have clarified it in the caption (Line 543)

Section 3.5. The title mentions implications for megacities, but the section only discusses lateral fluxes computed by the model. This doesn't add much to the paper. Either the discussion of relevance to megacities needs to be addressed in much more detail, which would turn this into a completely different paper. Or I would remove this and write a specific paper about this another time - which would be great because the representation of megacities in global hydrological modeling is a topic we should talk about more. Also showing the later fluxes makes for interesting maps but currently provides no scientific insights. Either this needs a direct comparison to deGraaf and Stahl (2022) and others, such as agupubs.onlinelibrary.wiley.com/doi/10.1029/2024WR038523 or I would remove this as well.

Response: We thank the reviewer for this constructive and inspiring comment. We agree that the discussion related to megacities was superficial in the current study. As suggested, we have now removed megacities from the section heading. We nonetheless kept a brief and clearly caveated discussion in the text because, as the reviewer suggested, the representation of megacities in global hydrological modeling is an emerging topic that needs to be addressed in future. Our intent is not to provide operational city-level assessments, but to use these examples as an exploratory illustration that the model may have potential for such global evaluations (Line 610 - 622).

Regarding the lateral groundwater flow analysis, we respectfully disagree that this component lacks scientific value. In fact, this section provides insightful global assessment of lateral groundwater fluxes simulated by a gradient-based steady-state MODFLOW model, which includes:

- Local OAT Sensitivity experiments revealing how lateral flux patterns and magnitudes are affected by key aquifer parameter setting (hydraulic conductivity), which was not reported in previous global groundwater studies.
- A qualitative intercomparison with recent global studies (e.g., de Graaf & Stahl, 2022); We have also included discussion relevant to the two recently published studies (Akhter et al. 2025 and Migueze-Macho and Fan 2025).

We agree that a more thorough analysis of lateral flows is necessary. However, given that the introduction of lateral flow was one of the core motivations behind extending H08 with a MODFLOW-based groundwater module, we believe that this section is an essential component of the current model development paper. We therefore position our current results as a starting point for further investigations into lateral groundwater processes in large-scale hydrological modeling in the future.

**Reviewer #2**

Thank you for this great contribution. The paper presents a global groundwater modeling framework (H08-GM) that couples the H08 global hydrological model with MODFLOW version 6 at 5 arcmin resolution. The authors focus on steady-state simulations under natural conditions. The study includes sensitivity analysis on aquifer parameters, validates simulated groundwater heads against Fan et al. (2013) water table depth dataset, and produces global maps of groundwater table depth and lateral flow.

This study addresses the growing demand for a better groundwater representation in global hydrological modelling. I see it as a valuable and well-executed newly developed model with parameter sensitivity analysis. It provides useful global visualizations and emphasizes the importance of improving subsurface data and lateral flow.

Overall, I recommend Minor revisions, mainly to improve clarity and figure presentation.

Response: We sincerely thank the reviewer for the positive feedback, which has encouraged us to further refine the work. All suggestions regarding manuscript clarity and figure presentation have been fully addressed in the revision.

**Specific comments**

The study is described as a coupled framework, but only a one-way coupling is implemented (H08 to MODFLOW). This limitation matters because groundwater feedback to surface processes is not represented. I suggest the authors be more explicit about this limitation and what it means for interpreting their results.

Response: Thank you for pointing out this important point. The current one-way coupling framework could cause an underestimation (deeper) of the groundwater head than it should be if the two-way simulation were enabled. The excessive groundwater is simply removed from the aquifer system, rather than entering to the surface water to strengthen its recharge to the aquifer. This could

cause underestimation of the simulated groundwater head (deeper groundwater levels) than it should be if the two-way simulation were enabled.

We have revised the methodological description in Section 2.1 to explicitly indicate this is a one-way coupled model (Line 104 - Line105), and added discussion of the possible consequences this may cause for the model simulation results in Conclusions (Line 615-619).

For the Aquifer thickness section (lines 196–210): I found this part hard to follow. The motivation for introducing aquifer thickness is not clearly connected to the model setup. Since the model simulates an unconfined aquifer, the authors should make the link between the aquifer thickness map, bottom elevation, and groundwater head more explicit.

Response: We apologize for the unclear description. The aquifer thickness refers to the depth between land surface elevation and aquifer bottom, and is a necessary input variable to MODFLOW. We have revised the diagram in Figure 1(b) to illustrate the relations between aquifer thickness, aquifer bottom elevation, and groundwater head. We have also revised the relevant text in Section 2.3.1 for a better clarification.

Figures could be improved with a few adjustments:

Many maps use red/blue scales that are not colorblind-friendly.

Response: We have adjusted all the maps with palette from Scientific colour maps 7.0, as seen in the revised manuscript.

b. Some captions are overly dense and read like mini-methods sections (e.g., Figure 2), while others don't give enough description or citations (e.g., Figure 3). Captions should primarily tell the reader what the figure shows; technical details can stay in the text.

Response: Thank you for your suggestion. We have revised the captions accordingly.

c. Units: Please make sure all colorbars explicitly show units.

Response: Thank you. We have added the colorbar labels for all figures.

The manuscript is too wordy in several places, with very long sentences that were difficult to follow. For example, lines 54–56, 85–90, 365–370. Breaking these into shorter sentences would make the paper easier to read.

Response: Thank you. We have revised them accordingly.

A few minor comments related to typos, clarity, and style (these are just examples, not a complete list):

Line 20: Typo — "hallow WTD" → "shallow WTD."

Response: We sincerely apologize for the typo. They are corrected now.

Line 28: "...compared to the two previous studies" — unclear which studies are meant; please name them explicitly.

Response: Thank you. As suggested by Referee #1, we have revised the sentence to include more global groundwater studies here.

Lines 54–56: Sentence too long. Suggest splitting into two: one on natural hydrology (supply), one on human use (demand).

Response: Thank you. The original sentence has been revised both linguistically and scientifically to fit a broader topic.

Line 67: ParFlow is also a groundwater model, not a land surface model.

Response: We have revised the sentence to highlight the coupling of ParFlow to CLM (Line 68).

Lines 85–90: Break into two sentences.

Response: Thank you. Revised (Line 70-74)

Lines 140–144: Too wordy and dense; hard to follow.

Response: We have revised them (Line 134-136).

Lines 175–176: "...monthly timestep ... aggregated to a monthly step" — I assume this is a typo; should be daily timestep aggregated to monthly.

Response: Thank you so much for pointing out this. The model is run at the monthly step. We have removed the latter sentence.

Lines 196-210 (Aguifer thickness): Hard to follow.

Response: We have revised this paragraph for a better clarity.

**Several inconsistent citations.**

Response: We have carefully proofread the revised manuscript to make sure all citations are consistent.

**Final comment**

This paper has strong potential. With clearer writing, stronger justification for the aquifer thickness part and the one-way limitation, and improved figures/captions, I believe it will be a valuable contribution to the global hydrology community.

Response: Again, we sincerely appreciate your encouraging comments and will refine our work to further advance our understanding on groundwater dynamics and their interactions with human activities.

**Reviewer #3**

This is an ambitious and timely global MODFLOW implementation that advances the representation of lateral groundwater flow. The manuscript is well organized and thoughtfully executed; addressing the points below would further strengthen its physical clarity and practical utility.

Response: We sincerely thank the reviewer for the positive comments, which have encouraged us to further refine the work. All suggestions have been fully addressed to improve the physical clarity and practical utility in the revised manuscript.

**Major comments**

H08 runs on a geographic grid, while MODFLOW uses a rectilinear grid in metric units. Please state if all areal and volumetric fluxes were converted.

Response:

We appreciate the reviewer's valuable comment. All the land surface variables in H08 relevant to MODFLOW model (e.g., recharge, runoff, etc.) are simulated at the flux density level (i.e., no grid cell area is involved). Therefore, we did not apply the areal and volumetric fluxes adjustment here. We have clarified this point in the revised manuscript (Lines 107-109).

Treating abstraction as a simple subtraction from net recharge neglects the spatial propagation of drawdown (cones of depression) and can bias heads even under steady state. Could you clarify this limitation? Do you intend to use the same approach in the transient runs? If so, wouldn't that undercut a key advantage of replacing a bucket model with an explicit groundwater model—namely, resolving spatially distributed drawdown and capture?

Response:

Thank you for this valuable comment. We would like to clarify two points here: (1) Our current steady-state run is under natural conditions without considering any pumping effect; and (2) For the transient run considering human pumping, we use MODFLOW's WEL package, rather than simply subtracting the pumping rate from recharge. The WEL package solves the flow field with point (or multi-node) sinks and

therefore produces the physically consistent drawdown cones. We will explicitly clarify this point in our upcoming manuscript. Thank you so much again for pointing out this issue.

Discuss biases expected where deep confined systems exist (e.g. North China Plain, Central Valley): vertical gradients, leakage from over/under-lying units, and coastal interfaces.

**Response:**

Thank you for this helpful suggestion. Omitting the water supply from underlying confined aquifer units and seawater intrusion in coastal areas can lead to an underestimation of the simulated groundwater head in our model. We have included a discussion of this limitation in the conclusion part (Lines 610 - 612).

For the net lateral groundwater flux, the current explanation is long and switches polarity to compare with other studies. consider adopt one sign convention throughout (and in captions) and move any polarity flips to the supplement.

Response:

Thank you for your comment. We have removed the description regarding the polarity switch to improve the manuscript clarity. We also condensed the relevant discussion regarding lateral flow, with only the most essential results retained to improve the overall clarity and readability of this section. See Section 3.5 in the revised manuscript.

Improve readability with a colorblind-safe palette. Consider annotating break values on the colorbar. Replace "aquifer conductivity" with "hydraulic conductivity (K)" throughout to avoid confusion.

**Response:**

We have adjusted all the colorbars with palette from Scientific colour maps 7.0, as seen in the revised manuscript. The terminology has also been revised throughout the manuscript

**Minor comments**

**L20: Typo hallow-> shallow**

Response: We apologize for the typo. It is now corrected.

L51–54: The list of global models is illustrative, not exhaustive. Rephrase to avoid implying exclusivity.

Response: We have revised the content accordingly (Line 46-50; Line 65-70).

L111–114: The absence of two-way coupling here stems from the steady-state design, not chiefly computational burden.

Response: We have revised it accordingly (Line 106-109).

L124–125: how to explain the reason behind? is it indicating that when the resolution is high enough, subgrid variation is not an important factor anymore? Response:

We appreciate the reviewer's valuable thoughts. Here we meant H08 shows good behavior in representing the global hydrological regime, i.e., soil moisture dominates evapotranspiration in arid areas, and net radiation dominates ET in humid areas. We have revised the sentence for a better clarity (Line 123-124).

**L141–142: Statement is unclear. I didn't understand.**

**Response:**

We meant the original H08 groundwater module is bucket-type and can only simulate groundwater storage changes (no level information). We have revised the sentence for a better clarity (Line 135-136).

L173: For near-surface air temperature downscaling, consider a lapse-rate correction, which is preferable to purely linear interpolation in high-relief regions. Response:

We appreciate this valuable suggestion. Implementing a lapse-rate correction would require reprocessing the meteorological forcing data and re-running the H08 simulations. Our current focus in this paper is on the groundwater module development. We will consider lapse-rate-based corrections for high-relief regions in

future work, especially when moving to transient, two-way H08–MODFLOW coupling.

L494: The claim about evaluating city-scale groundwater inflow/outflow from a global 5' (~10 km) model feels too strong. At this resolution—and given uncertainties in K, recharge, riverbed properties, and boundaries—such budgets are not robust. City-level assessments typically require carefully delineated regional/local models. Response:

We appreciate the reviewer's important clarification regarding the limitations. We agree that such budgets are not robust. In response, we removed "megacity" from the subsection title and toned down the claims.

We nonetheless kept a brief and clearly caveated discussion in the text because the representation of megacities in global hydrological modeling is an emerging topic that needs to be addressed in the future. Our intent is not to provide operational city-level assessments, but to use these examples as an exploratory illustration that the model may have potential for such global evaluations (Lines 586 - 589).

**Community Comments (Dr. Robert Reinecke)**

Dear Dr. Reinecke,

Thank you very much for your thoughtful comments and constructive suggestions. Please see our point-to-point reply below.

**On the color schemes:**

We sincerely apologize for the inappropriate palette choice. We have revised them accordingly based on Scientific Color Maps 7.0 for a more consistent and CVD-friendly visualization.

**On the literature you mentioned:**

We must admit that when we initiated this work, much of our time was invested in coding and in trying to better understand this relatively complex groundwater system. Your references on sensitivity analyses, spatial resolution effects, and uncertainty attribution in global groundwater modeling are highly relevant and very inspiring for our study. We have accommodated all of them and see them as an important foundation that will help us to interpret our model's current limitations more clearly. Please see the revised manuscript and our response to your referee's comments below

**On the ISIMIP groundwater sector:**

Thank you so much for your invitation. It is a truly an honor and valuable opportunity for us to join ISIMIP to learn from the ongoing efforts and to contribute our modeling results.

Thank you again for your valuable input, which has already helped us to strengthen our perspective.

**Community Comments #2 (Giacomo Medici)**

**Dear Giacomo Medici,**

We sincerely thank you for your careful reading of our manuscript and your constructive suggestions. We will carefully consider your comments regarding literature references, figure presentation, and writing structure, and revise the manuscript accordingly. See the revised manuscript and our point-to-point response below.

**Specific comments**

Lines 43-45. "Due to its large storage capacity and slow flow rate, groundwater contributes as the major and the most stable freshwater source to human water use in households, agriculture, and industry". Insert recent literature on storage and low flow rate in areas devolved to agriculture and industry:

- Medici, G., Munn, J.D., Parker, B.L. 2024. Delineating aquitard characteristics within a Silurian dolostone aquifer using high-density hydraulic head and fracture datasets. Hydrogeology Journal, 32(6), 1663-1691.
- Mukate, S.V., Panaskar, D.B., Wagh, V.M. and Baker, S.J., 2020. Understanding the influence of industrial and agricultural land uses on groundwater quality in semiarid region of Solapur, India. Environment, Development and Sustainability, 22(4), 3207-3238.

Response: Thank you for recommending these papers. We have included them in the revised manuscript (Line 39).

Lines 101. Disclose the overall aim / or goal of your research at the end of your introduction.

Response: We appreciate your comment. The overall aim of our research has already been clearly stated in the last paragraph of the Introduction (Lines 75–76). We believe this placement effectively summarizes the study's purpose and maintains

the logical flow of the section, so it does not necessarily need to appear at the very end.

Line 101. You need to describe the specify objectives of your research by using numbers (e.g., i, ii, and iii)

Response: We appreciate your suggestion to list the research objectives using numbered items. However, we believe the current paragraph already presents the objectives in a clear and logically connected manner. Therefore, we respectfully kept the narrative form to maintain the flow and coherence of the text.

**Line 105. Do you need to specify a MODFLOW version?**

Response: We believe that specifying the exact MODFLOW version is important for clarity and reproducibility, as different versions may use different numerical schemes and packages. Therefore, we have retained the version information in the manuscript.

Line 363. What about ME, MAE, RMS and R2 for the difference between model and observation?

Response: Thank you for your suggestion. We have indicated additional statistics in the newly added scatterplots (Figure 6 and Figure 7) and global validation map (Figure 8)

Lines 527-528. There should be no references in a conclusion.

Response: We appreciate your comment. However, we respectfully note that including references in the conclusion is acceptable in scientific writing, particularly when summarizing findings in the context of previous studies. We have therefore retained a few key citations to highlight the broader relevance of our results.

**Figures and tables**

Figure 1a, b. Some words are unreadable. Please, make them larger and increase the graphic resolution.

Response: Thank you. We have adjusted them accordingly.

**Figure 1a. Specify 2D diagram with a single layer?**

Response: We have clarified that the illustration represents a single-layer aquifer (Line 129); however, the diagram itself is shown in 3D for better visualization.

**Figure 1a. Specify structured grid?**

Response: Thanks. Added (Line 129).

**Figure 2. The three figures on the bottom can be closer and then can be enlarged.**

Response: We appreciate your suggestion. We have carefully adjusted the layout of Figure 2 to improve spacing and readability. However, due to the limitations of the plotting library (Matplotlib) and the need to maintain consistent axis scales and colorbars across subplots, the current layout in the revised manuscript represents the best achievable configuration without compromising figure clarity.

**Figure 3. Better "Aquifer Hydraulic Conductivity"**

Response: Thanks. Adjusted.

**Figure 6. Some issues here. The legends cover some parts of the figures and some details are un-readable.**

Response: We have tested several alternative layouts to improve readability, but due to scaling and formatting constraints, the current configuration is the best achievable version. The legends are intentionally placed over regions with missing or sparse data, so no meaningful details are obscured.

Once again, we highly appreciate your time and effort in helping us improve our work.

**Development of A Global 5arcmin Groundwater Model (H08-GMv1.0): Model Setup and Steady-State Simulation**

Qing He1, Naota Hanasaki1,2, Akiko Matsumura3, Edwin H. Sutanudjaja4, Taikan Oki1

Correspondence to: Qing He (heq16@tsinghua.org.cn)

**Abstract.** Groundwater plays a critical role in regulating the global hydrological cycle and serves as the most stable freshwater resource for human daily water consumption. However, many global water models, including H08, a global water model considering human water use activities, downplay the groundwater component, i.e., the underground aquifer is often described as a simple lumped model where no lateral groundwater movement or the water table is represented. Here, we present a global H08-MODFLOW groundwater model (H08-GM), built at a five-arcmin spatial resolution, aiming to enhance the capability of the original H08 model in simulating groundwater flows. We describe the basic model setups and simulations under steadystate conditions in this paper. The Local One-At-A-Time (OAT) Sensitivity Tests are first conducted to select the best-run model simulations against in-situ observations. At the global scale, all model runs demonstrate overall good performance of groundwater head, whereas perform poorly in simulating Water Table Depth (WTD, groundwater table below land surface), which is shown to be a common issue in other global groundwater models. However, the model's WTD behaviour is reasonably well in densely populated and irrigated areas, demonstrating its validity for application relevant to human water use activities. We further use the model to reveal the mechanisms controlling groundwater flow dynamics and present the global cell-to-cell net groundwater lateral flow map. We found that the magnitude in some regions is non-negligible to annual groundwater recharge. This highlights the important role of the lateral groundwater flow in maintaining the regional water budget. The steady-state simulation from this study provides the necessary initial condition for the transient simulations, which is essentially important to analyze the global groundwater decline trends and will be presented in another paper. Although developed in the one-way coupled manner, the H08-GM model can provide a powerful tool for large-scale groundwater studies, which enables direct comparison with other groundwater models joined the Inter-Sectoral Impact Model Intercomparison Project (ISIMIP), and is essential to advance the development of the next-generation global water models.

**1 Introduction**

Groundwater plays a critical role in the global hydrological cycle. The water exchange between aquifers and surface water bodies buffers the sharp seasonal fluctuations in river channels and lakes, maintaining the resilience of aquatic landscapes and

<sup>1Department of Civil Engineering, The University of Tokyo, Tokyo, 113-8656, Japan

<sup>2National Institute for Environmental Sciences, Tsukuba, 305-8506, Japan

<sup>3Nippon Koei Co. Ltd., Tsukuba, 305-0047, Japan

[revised manuscript text omitted]

 to the previous research in de Graaf et al. (2015), which includes approximately 1000 Monte Carlo sensitivity experiments, the limited sensitivity analyses in our study may be subject to the degraded confidence of the selected optimal parameter settings. However, the fact that the simulated groundwater heads compare reasonably well to the in-situ observations globally confirms the feasibility of our model, although more comprehensive parameter tunings are suggested in the future.

635

645

Our model contributes as one of the three major GHWMs that explicitly considers groundwater lateral flow at the global scale. Additionally, the capability of H08-GM to directly output groundwater levels, calculate lateral flow rate, and connect rivers and aquifers, provides a powerful tool to investigate the groundwater decline trend over the pumping hotspots in the world, to identify river basins as importers or exporters, and to examine the losing and gaining regimes of streamflow. It will essentially help improve the accuracy of the water resource availability estimated based on the original H08 model. The steady-state simulation result in this paper has demonstrated the 40-year mean natural groundwater level distribution without human disturbance. We will show the temporal groundwater level variability and the human water withdrawal effect over the past 40 years in a following paper, which will help further advance our understanding of the important role of groundwater in supporting human water consumption, and the fundamental mechanisms behind the human-groundwater interactions.

**Appendix A: Algorithms to calculate river channel depth and river width**

In the latest version of CaMaFlood, the river channel depth  $(D_{chn})$  is calculated based on the power-law empirical equation, as:

$$D_{chn} = max (H_{min}, H_c * Q_{chn}^{H_p} + H_0)$$
 (A1)

Where,  $H_{min}$  = 1.0 is the prescribed minimum channel depth (unit: m);  $H_c = 0.1$  and  $H_p = 0.50$  are the coefficients,  $H_0 = 0.00$  is the prescribed offset number for river channels;  $Q_{chn}$  is the river discharge (unit: m3 s-1).

The river width  $(RIV_{wth})$  is obtained based on both satellite observation and power-law estimation. The satellite-derived river width is first read in as the baseline variable  $(RIV_{gwalr})$ . The river width based on power-law  $(RIV_{wth})$  is then calculated separately, as:

$$RIV_{wth} = max \left( W_{min}, W_c * Q_{chn}^{W_p} + W_0 \right) \tag{A2}$$

Where,  $W_{min} = 5.0$  is the prescribed minimum river channel width (unit: m),  $W_c = 2.50$  and  $W_p = 0.60$  are the coefficients, and  $W_0 = 0.00$  is the prescribed offset number;  $Q_{chn}$  is the river discharge (unit: m3/s).

Afterwards,  $RIV_{wth}$  is used to constrain the underestimation of  $RIV_{gwdlr}$  for small rivers and overestimation for large rivers, as:

$$RIV_{gwdlr} = \begin{cases} \max(RIV_{gwdlr}, RIV_{wth}), & if \ RIV_{gwdlr} < 50 \\ RIV_{wth} * 0.5, & if \ RIV_{gwdlr} < RIV_{wth} * 0.5 \\ RIV_{wth} * 5.0, & if \ RIV_{gwdlr} > RIV_{wth} * 5.0 \\ 10000, & if \ RIV_{gwdlr} > 10000 \end{cases} \tag{A3}$$

Appendix B. Look-up table for lithology-based aquifer conductivity (log-transformed, m²).

sample size, and standard deviation  $(\sigma)$  for each subcategory are listed. The category-averaged mean  $(\overline{\mu_{log K}})$  and standard deviation  $(\overline{\sigma_{log K}})$  are then The 16 lithology categories are PY (Pycroclastics), VB (Basic Volcanic Rocks), PA (Acid Plutonic Rocks), MT (Metamorphic Rocks), SU (Unconsolidated Sediments), SS (Siliciclastic Sedimentary Rocks), ND (No Data), PB (Basic Plutonic Rocks), SM (Mixed Sedimentary Rocks), WB (Water Bodies), VI (Intermediate Volcanic Rocks), SC (Carbonate Sedimentary Rocks), VA (Acid Vocanic Rocks), EV (Evaporites), PI (Intermediate Plutonic Rocks), IG (Ice and Glaciers). For each lithological categories there are maximum 4 subcategories (SbC). The mean ( $\mu$ ), calculated and given in the last two columns.

| G (m 2 )            | Logk (III)                               | 2.0   | 1.8   | 1.5   | 1.5   | 1.8   | 2.1   | 0.0   | 1.5   | 2.1   | 0.0   | 1.8   |
|--------------------------------|------------------------------------------|-------|-------|-------|-------|-------|-------|-------|-------|-------|-------|-------|
| // (m 2 )           | $\overline{\mu_{log K}}  (\mathrm{m}^2)$ |       | -12.5 | -14.1 | -14.1 | -12.6 | -15.0 | -20.0 | -14.1 | -15.0 | -20.0 | -12.5 |
|                                | SbC4                                     | 0.0   | 0.0   | 0.0   | 0.0   | 2.0   | 2.5   | N/A   | 0.0   | 2.5   | N/A   | 0.0   |
| (m 2 )              | SbC3                                     | 1.8   | 0.0   | 0.0   | 0.0   | 1.8   | 1.7   | N/A   | 0.0   | 1.7   | N/A   | 0.0   |
| $\sigma_{logK}~(\mathrm{m}^2)$ | SbC2                                     | 2.5   | 1.8   | 1.5   | 1.5   | 1.2   | 6.0   | N/A   | 1.5   | 6.0   | N/A   | 0.0   |
|                                | SbC1                                     | 2.0   | 1.8   | 1.5   | 1.5   | 2.0   | 2.5   | N/A   | 1.5   | 2.5   | N/A   | 1.8   |
|                                | SbC4                                     | 0     | 0     | 0     | 0     | 113   | 20    | N/A   | 0     | 20    | N/A   | 0     |
| size (-)                       | SbC3                                     | 33    | 0     | 0     | 0     | 31    | 11    | N/A   | 0     | 11    | N/A   | 0     |
| Sample size (-)                | SbC2                                     | 20    | 33    | 17    | 17    | 82    | 6     | N/A   | 17    | 6     | N/A   | 0     |
|                                | SbC1                                     | 113   | 33    | 17    | 17    | 113   | 20    | N/A   | 17    | 20    | N/A   | 33    |
|                                | SbC4                                     | 0.0   | 0.0   | 0.0   | 0.0   | -13.0 | -15.2 | N/A   | 0.0   | -15.2 | N/A   | 0.0   |
| (m 2 )              | SbC3                                     | -12.5 | 0.0   | 0.0   | 0.0   | -14.0 | -16.5 | N/A   | 0.0   | -16.5 | N/A   | 0.0   |
| µ logK              | SbC2                                     | -15.2 | -12.5 | -14.1 | -14.1 | -10.9 | -12.5 | N/A   | -14.1 | -12.5 | N/A   | 0.0   |
|                                | SbC1                                     | -13.0 | -12.5 | -14.1 | -14.1 | -13.0 | -15.2 | N/A   | -14.1 | -15.2 | N/A   | -12.5 |
| Cat                            |                                          | PY    | VB    | PA    | MT    | SU    | SS    | ND    | PB    | SM    | WB    | VI    |

(continued)

[revised manuscript text omitted]

---

## Referee Report (RR1)

Thank you for the revised version. The authors have addressed the previous comments thoroughly, and the new draft is well detailed and clearly presented.
I have small comments, please clarify what "GHWMs" in line 60 refers to, and the citation in line 46 has to be at the end of sentence.

---

## Author Response (AR2)

Dear Dr. Cenlin He,

Thank you so much again for handling our manuscript. Below, we respond to the three reviewers' feedback. We have improved our manuscript in the several aspects below:

- We have revised the wording of the model's validation on WTD performance and discussions in the conclusion part;
- We have clarified several technical details of the model performance in terms of human gradients.

By incorporating these improvements, we hope that we have satisfactorily addressed all reviewer comments and that the revised manuscript will meet the standards for publication in GMD

Below, we provide our responses in blue text, while the reviewers' comments remain as black.

Sincerely regards,

Qing He

on behalf of all coauthors

**Reviewer #1 (Dr. Robert Reinecke)**

I applaud the authors for a comprehensive revision that addressed all my previous remarks. However, I also have a couple of new remarks that should be addressed before this can be accepted for publication.

Response:

     Dear Dr. Reinecke, we sincerely thank you for the positive feedback, and we are grateful for the additional remarks provided in this round. We have carefully addressed each new comment point by point as below.

The abstract now states, "However, the model's WTD behaviour is reasonably well in densely populated and irrigated areas, demonstrating its validity for application relevant to human water use activities".

I would disagree with that statement. Which part of your analysis supports this conclusion? Also, what does reasonably well mean? Please provide a quantification.

Response:

     We thank the reviewer for this valuable comment and agree that the previous conclusion was overstated. See our response to your next comment below.

     We have revised the abstract to remove any subjective assessment of model performance on WTD. The sentence now reads:

     "Our analysis also reveals two complementary global relationships: one between groundwater depth and topographic slope, and another along gradients of human activity (irrigation and population), together demonstrating how natural and anthropogenic processes jointly control the spatial distribution of WTD." (Line 18-20)

I have the same concern about the description of Fig. 10. While it is interesting to see that the general spread of WTD is lower in highly irrigated and populated areas, this does not necessarily mean it is sufficient. Just because the mean is similarly close to a very uncertain ensemble is not enough. Also, even if the mean is close to the observed mean, it does not tell us anything about the model's validity in specific places or in general. I suggest removing this part of the analysis.

Response:

     Thank you for this insightful comment and we fully agree that our earlier statement based solely on the comparison of mean values in the boxplots may have overstated the

model performance. To better illustrate the information not captured by the boxplots, we have further added stratified histograms (below and in the revised Fig. 10). The results show that even if highly irrigated and populated regions exhibit a narrower spread of WTD biases, the number of samples with small residuals (low bias) also increases across both low and high human-influence groups, which confirms your concerns.

[Figure]

However, we prefer to retain this part of the analysis, as it does not aim to assess the model's validity at specific local scales, but rather reveals a systematic and interpretable relationship between human activities and the model's WTD biases at the global scale. The analysis is similar to the previously discussed WTD-slope relationship, but complement it with a focus on anthropogenic processes. We believe the analysis here is still valuable to provide insights into the model's performance under different climatic and human activity gradients, as Reviewer #3 already pointed out. A comprehensive site-level validation of model performance would indeed be valuable, and we intend to further explore this aspect in our future work.

Following your suggestion, we have revised the corresponding text in the manuscript to clearly state this intent and to avoid any implication that the model performance in these regions is "sufficient". (Line 494 - 500)

Line 21: magnitude of what?

Response: It is "the magnitude of the net groundwater lateral flow". Corrected now. (Line 22)

Fig. 6: All figures should contain a complete, standalone description that enables the reader to understand the figure.

Response: Thank you for this comment. We have added detailed captions in the revised manuscript (Line 417 - 420).

Fig. 11: Does this need to be in the main manuscript? I can't make out any differences on these small maps. I suggest moving this to the supplement.

Response: Thank you. We have moved it to the Appendix now.

620: de Graaf (2015) evaluated mainly the coefficient of variation of the model output as a sensitivity analysis. I disagree that this provided more insights than your OAT experiment. In Reinecke 2019 HESS, we provide a much more comprehensive analysis. I suggest citing both studies here.

Response: We sincerely thank the reviewer for this insightful comment and for recognizing the value of our sensitivity analysis. We have now cited both de Graaf et al. (2015) and Reinecke et al. (2019) and revised the text accordingly. The text now reads as:

"Compared with the earlier global sensitivity analysis by de Graaf et al. (2015), which mainly evaluated the coefficient of variation of model outputs, and the more comprehensive subsequent study by Reinecke et al. (2019b), which systematically quantified model sensitivity to both individual and combined parameter variations through an extensive set of 1,848 Monte Carlo experiments, our OAT sensitivity test provides a complementary but more limited perspective on parameter uncertainty." (Line 665 - 675)

631 and following: this needs to be rephrased. Also, avoid making a specific promise about a future paper in the conclusions. That should only be made if this is a multi-part paper from the beginning. I suggest a general remark on future research here.

Response: Thank you for your suggestion. We have revised the context accordingly. The text now reads as:

"In the next step, the temporal groundwater level variability and the human water withdrawal effect over the past 40 years should be investigated to help further advance our understanding of the important role of groundwater in supporting human water

consumption, and the fundamental mechanisms behind the human-groundwater interactions." (Line 685 - 687)

**Reviewer #2:**

Thank you for the revised version. The authors have addressed the previous comments thoroughly, and the new draft is well detailed and clearly presented.I have small comments, please clarify what "GHWMs" in line 60 refers to, and the citation in line 46has to be at the end of sentence.

Response: We sincerely thank the reviewer for the positive feedback and further comments. The term "GHWMs" in line 60 of the original manuscript was left from an earlier draft and has now been corrected to "GWMs" throughout the revised version for consistency. We have also moved the citation in Line 46 of the original manuscript to the end of the sentence in the revised manuscript (Line 49 and Line 64).

**Reviewer #3:**

I would like to thank the authors for their thorough and thoughtful revision. The manuscript has been substantially improved in clarity, structure, and scientific depth. Most of my previous comments have been carefully addressed, and I am overall satisfied with the revision. The inclusion of the additional validation in Section 3.2 is particularly valuable and provides new insights into the model's performance under different climatic and human activity gradients.

Though I have two remaining points regarding the new Section 3.2.

Response: Dear Reviewer, thank you so much for your positive comments and we are glad to know our last round revision has met your expectation. Please see below for our point-to-point response to your new remarks.

Figure 10: The WTD boxplots for H08-GM show much wider ranges (25–75%) than those of the ensemble mean. Could the authors briefly explain why this spread is so large? For example, is it because the model captures more variability from the H08 forcing and lithology parameters, or because of larger uncertainty from the single-layer steady-state setup?

Response:

   We thank the reviewer for this thoughtful question. The wider interquartile range (25–75%) of WTD in H08-GM compared to the ensemble mean can be partly explained by the averaging nature of the ensemble. Since the ensemble mean combines the outputs from four different global groundwater models, two of which (Fan et al., 2013; Reinecke et al., 2019) produce systematically shallower WTD (see Fig. 2 in Reinecke et al., 2024), the averaging process inherently smooths spatial variability and reduces the spread.

On the other hand, the H08-GM simulations retain more of the spatial heterogeneity arising from its specific forcing data, lithological properties, and parameterization, as the reviewer pointed out. Identifying which factors dominate this difference would require coordinated experiments under consistent simulation settings across all models, which we consider an important next step for future intercomparison work.

We have added the above discussion in the revised manuscript as well (Line 485 - 491.

Figure 9: The slope–WTD relationships look clear, but it would be great to add a simple number (like a correlation R or R²) to quantify how well the model agrees with the observations and ensemble mean.

Response:

We thank the reviewer for this constructive suggestion. Following the advice, we have now added the Spearman correlation coefficients ($\rho$) and their significance levels (*** for $p < 0.001$) in each panel of the revised figure, as the relationships should be non-linear.

The results show that for observations (left column), the correlation between slope and WTD is generally weak ($\rho < 0.2$) under both energy-limited and water-limited conditions. In contrast, the ensemble mean exhibits much stronger correlations ($\rho \approx 0.6$), suggesting that the multi-model mean tends to emphasize a stronger slope–WTD dependence. The Spearman $\rho$ of H08 are closer to those of the observations; however, it should be noted that the relatively low numerical correlations may partly result from the large variability of WTD within each slope bin. When looking at the medians, a stronger relationship with slope can still be observed.

This pattern therefore suggests that current global numerical groundwater models may all tend to overemphasize the "groundwater head follows topography" relationship; Or in other words, they potentially underrepresent the influence of other factors such as climate forcing and local aquifer properties.

We have added the relevant discussion in the revised manuscript (Line 464 - 472).

[Figure]

Overall, the revised manuscript is close to publication quality.

Response: Once again, we sincerely thank you for your valuable and constructive comments, which greatly help us improve the manuscript.